# Analysis of Classifier-Free Guidance Weight Schedulers

**Xi Wang**[1], **Nicolas Dufour**[1,2], **Nefeli Andreou**[3†], **Marie-Paule Cani**[1], **Victoria Fernández Abrevaya**[4], **David Picard**[2*], **Vicky Kalogeiton**[1*]

[1] *LIX, École Polytechnique, CNRS, IPP*  *firstname.lastname@polytechnique.edu*
[2] *LIGM, École des Ponts, Univ Gustave Eiffel, CNRS*  *firstname.lastname@enpc.fr*
[3] *University of Cyprus*  *nefeliandreou@outlook.com*
[4] *Max Planck Institute for Intelligent Systems, Tübingen*  *victoria.abrevaya@tuebingen.mpg.de*
[†] *Work done during an internship at LIX, prior to joining Amazon.*  [*] *Denotes equal supervision*

**Reviewed on OpenReview:** *https://openreview.net/forum?id=SUMtDJqicd*

## Abstract

Classifier-Free Guidance (CFG) enhances the quality and condition adherence of text-to-image diffusion models. It operates by combining the conditional and unconditional predictions using a fixed weight. However, recent works vary the weights throughout the diffusion process, reporting superior results but without providing any rationale or analysis. By conducting comprehensive experiments, this paper provides insights into CFG weight schedulers. Our findings suggest that simple, monotonically increasing weight schedulers consistently lead to improved performances, requiring merely a single line of code. In addition, more complex parametrized schedulers can be optimized for further improvement, but do not generalize across different models and tasks.

## 1 Introduction

Diffusion models have demonstrated prominent generative capabilities in various domains e.g. images (Ho et al., 2020), videos (Luo et al., 2023), acoustic signals (Kang et al., 2023b), or 3D avatars (Chen et al., 2023). Conditional generation with diffusion (e.g. text-conditioned image generation) has been explored in numerous works (Saharia et al., 2022; Ruiz et al., 2023; Balaji et al., 2022), and is achieved in its simplest form by adding an extra condition input to the model (Nichol & Dhariwal, 2021). To increase the influence of the condition on the generation process, Classifier Guidance (Dhariwal & Nichol, 2021) proposes to linearly combine the gradients of a separately trained image classifier with those of a diffusion model. Alternatively, Classifier-Free Guidance (CFG) (Ho & Salimans, 2021) simultaneously trains conditional and unconditional models, and exploits a Bayesian implicit classifier to condition the generation without an external classifier.

In both cases, a weighting parameter $\omega$ controls the importance of the generative and guidance terms and is directly applied at all timesteps. Varying $\omega$ is a trade-off between fidelity and condition reliance, as an increase in condition reliance often results in a decline in both fidelity and diversity. In some recent literature, the concept of dynamic guidance instead of constant one has been mentioned: MUSE (Chang et al., 2023) observed that a linearly increasing guidance weight could enhance performance and potentially increase diversity. This approach has been adopted in subsequent works, such as in Stable Video Diffusion (Blattmann et al., 2023), and further mentioned in Gao et al. (2023) through an exhaustive search for a parameterized cosine-based curve (pcs4) that performs very well on a specific pair of model and task. Intriguingly, despite the recent appearance of this topic in the literature, none of the referenced studies has conducted any empirical experiments or analyses to substantiate the use of a guidance weight scheduler. For instance, the concept of linear guidance is briefly mentioned in MUSE (Chang et al., 2023), around Eq. 1: *"we reduce the hit to diversity by linearly increasing the guidance scale t [...] allowing early tokens to be sampled more freely"*. Similarly, the pcs4 approach Gao et al. (2023) is only briefly discussed in the appendix, without any detailed ablation or comparison to static guidance baselines. Thus, to the best of our knowledge, a comprehensive guide to dynamic guidance weight schedulers does not exist at the moment.

In this paper, we bridge this gap by delving into the behavior of guidance and systematically examining its influence on the generation, discussing the mechanism behind dynamic schedulers and the rationale for their enhancement. We explore various heuristic dynamic schedulers and present a comprehensive benchmark of both heuristic and parameterized dynamic schedulers across different tasks, focusing on fidelity, diversity, and textual adherence. Our analysis is supported by quantitative, and qualitative results and user studies.

Low static guidance:

```
w = 2.0
for t in range(1, T):
  eps_c = model(x, T-t, c)
  eps_u = model(x, T-t, 0)
  eps = (w+1)*eps_c - w*eps_u
  x = denoise(x, eps, T-t)
```

✗ Fuzzy images, but many details and textures

High static guidance:

```
w = 14.0
for t in range(1, T):
  eps_c = model(x, T-t, c)
  eps_u = model(x, T-t, 0)
  eps = (w+1)*eps_c - w*eps_u
  x = denoise(x, eps, T-t)
```

✗ Sharp images, but lack of details and solid colors

Dynamic guidance:

```
w0 = 14.0
for t in range(1, T):
  eps_c = model(x, T-t, c)
  eps_u = model(x, T-t, 0)
  # clamp-linear scheduler
  w = max(1, w0*2*t/T)
  eps = (w+1)*eps_c - w*eps_u
  x = denoise(x, eps, T-t)
```

✓ Sharp images with many details and textures, without extra cost.

"full body, a cat dressed as a Viking, with weapons in his paws, on a Viking ship, battle coloring, glow hyper-detail, hyper-realism, cinematic, trending on artstation"

Figure 1: Classifier-Free Guidance introduces a trade-off between detailed but fuzzy images (low guidance, top) and sharp but simplistic images (high guidance, middle). Using a guidance scheduler (bottom) is simple yet very effective in improving this trade-off.

Our findings are the following: First, we show that too much guidance at the beginning of the denoising process is harmful and that monotonically increasing guidance schedulers are performing the best. Second, we show that a simple linearly increasing scheduler always improves the results over the basic static guidance, while costing no additional computational cost, requiring no additional tuning, and being extremely simple to implement. Third, a parameterized scheduler, like clamping a linear scheduler below a carefully chosen threshold (Figure 1), can significantly further improve the results, but the choice of the optimal parameter does not generalize across models and tasks and has thus to be carefully tuned for the target model and task. All our findings are guides to CFG schedulers that will benefit and improve all works relying on CFG.

## 2 Related Work

**Generative and Diffusion Models.** Before the advent of diffusion models, several generative models were developed to create new data that mimics a given dataset, either unconditionally or with conditional guidance. Notable achievements include Variational AutoEncoders (VAEs) (Kingma & Welling, 2014) and

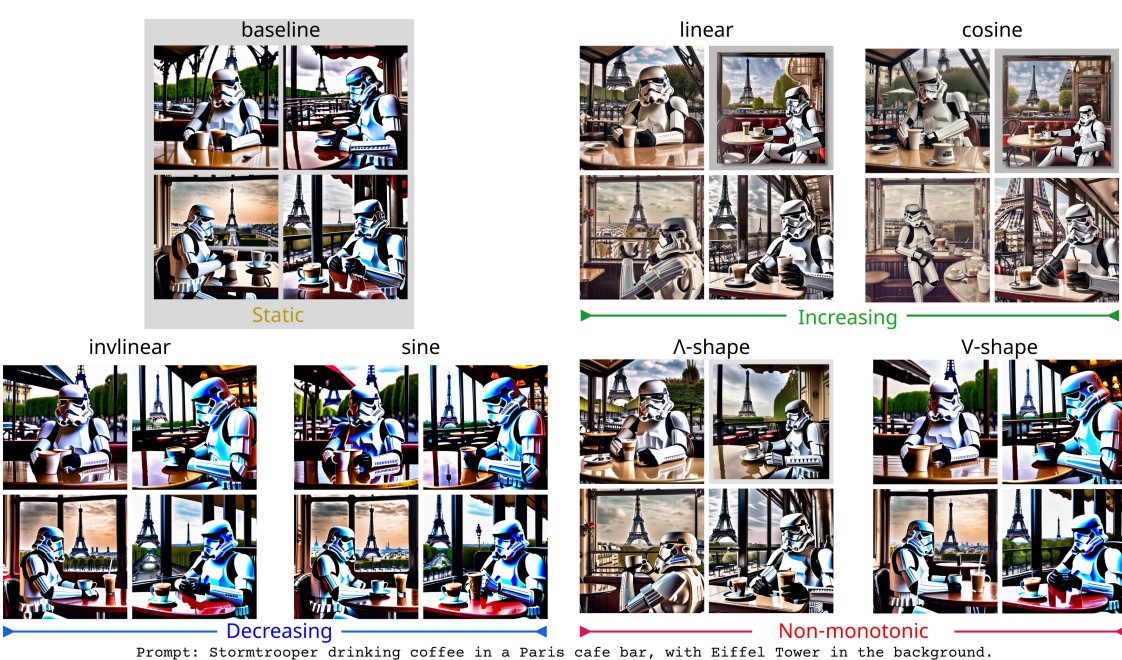

Figure 2: **Examples of all heuristics** on SDXL. Increasing ones (*linear* and *cosine*) enhance fidelity, textual adherence and diversity.

Generative Adversarial Networks (GANs) (Goodfellow et al., 2014), which have recorded significant progress in various generative tasks (Brock et al., 2018; Kang et al., 2023a; Dufour et al., 2022; Donahue et al., 2018). Recently, diffusion models have demonstrated a remarkable capacity to produce high-quality and diverse samples. They have achieved state-of-the-art results in several generation tasks, notably in image synthesis (Song et al., 2020; Ho et al., 2020), text-to-image applications (Dhariwal & Nichol, 2021; Rombach et al., 2022; Podell et al., 2023; Pernias et al., 2023) and text-to-motion (Chen et al., 2023).

**Guidance in Diffusion and Text-to-Image.** Making generative models controllable and capable of producing user-aligned outputs requires making the generation conditional on a given input. Conditioned diffusion models have been vastly explored (Saharia et al., 2022; Ruiz et al., 2023; Balaji et al., 2022). The condition is achieved in its simplest form by adding extra input, typically with residual connections (Nichol & Dhariwal, 2021). To reinforce the model's fidelity to specific conditions, two main approaches prevail: Classifier Guidance (CG) (Dhariwal & Nichol, 2021), which involves training an image classifier externally, and Classifier-Free Guidance (CFG) (Ho & Salimans, 2021), that relies on an implicit classifier through joint training of conditional and unconditional models (using dropout on the condition).

Particularly, CFG has catalyzed advancements in text-conditional generation, a domain where training a noisy text classifier is less convenient and performs worse. This approach breathed new life into the text-to-image application, initially proposed in several works such as (Reed et al., 2016; Mansimov et al., 2015). Numerous works (Rombach et al., 2022; Ramesh et al., 2022; Nichol et al., 2022; Avrahami et al., 2022) have leveraged text-to-image generation with CFG diffusion models conditioned on text encoders like CLIP (Radford et al., 2021), showcasing significant progress in the field, e.g. the Latent Diffusion Model (Dhariwal & Nichol, 2021) and Stable Diffusion (Rombach et al., 2022) employ VAE latent space diffusion with CFG with CLIP encoder. SDXL, an enhanced version, leverages a larger model and an additional text encoder for high-resolution synthesis.

**Improvements on Diffusion Guidance.** Noticed that in *Classifier Guidance (CG)*, the classifier's gradient tends to vanish towards the early and final stages due to overconfidence, Zheng et al. (2022) leverages the entropy of the output distribution as an indication of vanishing gradient and rescales the gradient accordingly. To prevent such adversarial behaviours, Dinh et al. (2023b) explored using multiple class conditions, guiding the image generation from a noise state towards an average of image classes before focusing on the

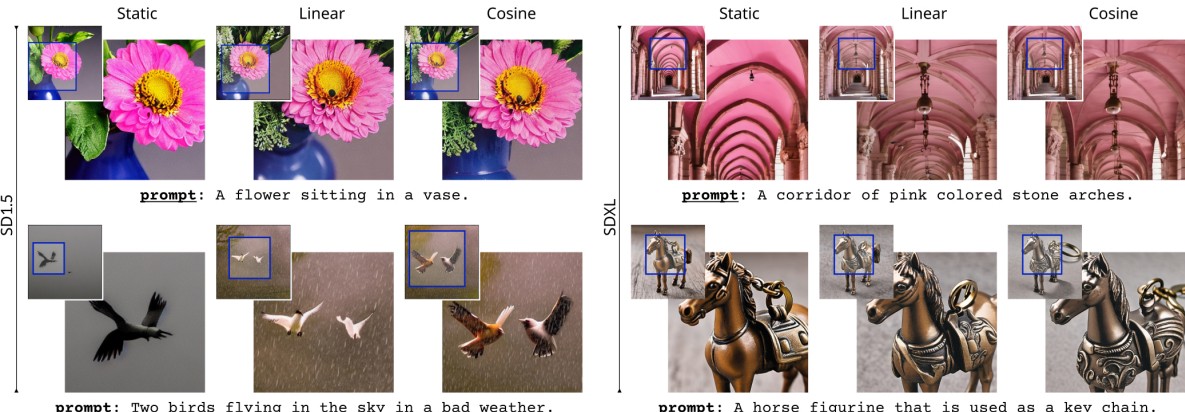

Figure 3: **Qualitative results of fidelity** for different guidance schedulers compared with static baseline. *linear* and *cosine* schedulers show better image details (flower petal, figurine engraving), more natural colour (pink corridor), and better textual adherence (bad weather for the two birds image, key-chain of the figurine).

desired class with an empirical scheduler. Subsequently, Dinh et al. (2023a) identified and quantified gradient conflicts emerging from the guidance and suggested gradient projection as a solution.

In *Classifier-Free Guidance (CFG)*, Li et al. (2023) used CFG to recover a zero-shot classifier by sampling across timesteps and averaging the guidance magnitude for different labels, with the lowest magnitude corresponding to the most probable label. However, they observed a discrepancy in performance across timesteps with early stages yielding lower accuracy than intermediate ones. Chang et al. (2023) observed that a linear increase in guidance scale enhances diversity. Similarly, Gao et al. (2023) developed a parameterized power-cosine-like curve, optimizing a specific parameter for their dataset and method. However, these linear and power-cosine schedulers have been suggested as improvements over constant static guidance without rigorous analysis or testing. To this end, we provide an extensive study of dynamic guidance for both heuristic and parametrized schedulers across several tasks. Concurrently, Kynkäänniemi et al. (2024) proposes empirically removing the initial and final timesteps of the classifier-free guidance (CFG) for improved generation. Similarly, Zhang et al. (2024); Castillo et al. (2023) observes that the conditional and unconditional responses of some models may converge to similar behaviours at certain timesteps, particularly towards the ending stage.

## 3  Background on Guidance

Following DDPM (Ho et al., 2020), diffusion consists in training a network $\epsilon_\theta$ to denoise a noisy input to recover the original data at different noise levels. More formally, the goal is to recover $x_0$, the original image from $x_t = \sqrt{\gamma(t)}x_0 + \sqrt{1-\gamma(t)}\epsilon$, where $\gamma(t) \in [0,1]$ is a noise scheduler of the timestep $t$ and applied to a standard Gaussian noise $\epsilon \sim \mathcal{N}(0,1)$. In practice, Ho et al. (2020) find that predicting the noise $\epsilon_\theta$ instead of $x_0$ yielded better performance leading to the training loss: $L_{\text{simple}} = \mathbb{E}_{x_0 \sim p_{\text{data}}, \epsilon \sim \mathcal{N}(0,1), t \sim \mathcal{U}[0,1]} \left[ \|\epsilon_\theta(x_t) - \epsilon\| \right]$ based on the target image distribution $p_{\text{data}}$ with $\mathcal{U}$ uniform distributions.

Once the network is trained, we can sample from $p_{\text{data}}$ by setting $x_T = \epsilon \sim \mathcal{N}(0,1)$ (with $\gamma(T)=0$), and gradually denoising to reach the data point $x_0 \sim p_{\text{data}}$ with different types of samplers e.g., DDPM (Ho et al., 2020) or DDIM (Song et al., 2020). To leverage a condition $c$ and instead sample from $p(x_t|c)$, Dhariwal & Nichol (2021) propose *Classifier Guidance (CG)* that uses a pretrained classifier $p(c|x_t)$, forming:

$$\nabla_{x_t} \log p(x_t|c) = \nabla_{x_t} \log p(x_t) + \nabla_{x_t} \log p(c|x_t) \quad , \tag{1}$$

according to Bayes rule. This leads to the following *Classifier Guidance* equation, with a scalar $\omega > 0$ controlling the amount of guidance towards the condition $c$:

$$\hat{\epsilon}_\theta(x_t, c) = \epsilon_\theta(x_t) + (\omega + 1)\nabla_{x_t} \log p(c|x_t) \quad . \tag{2}$$

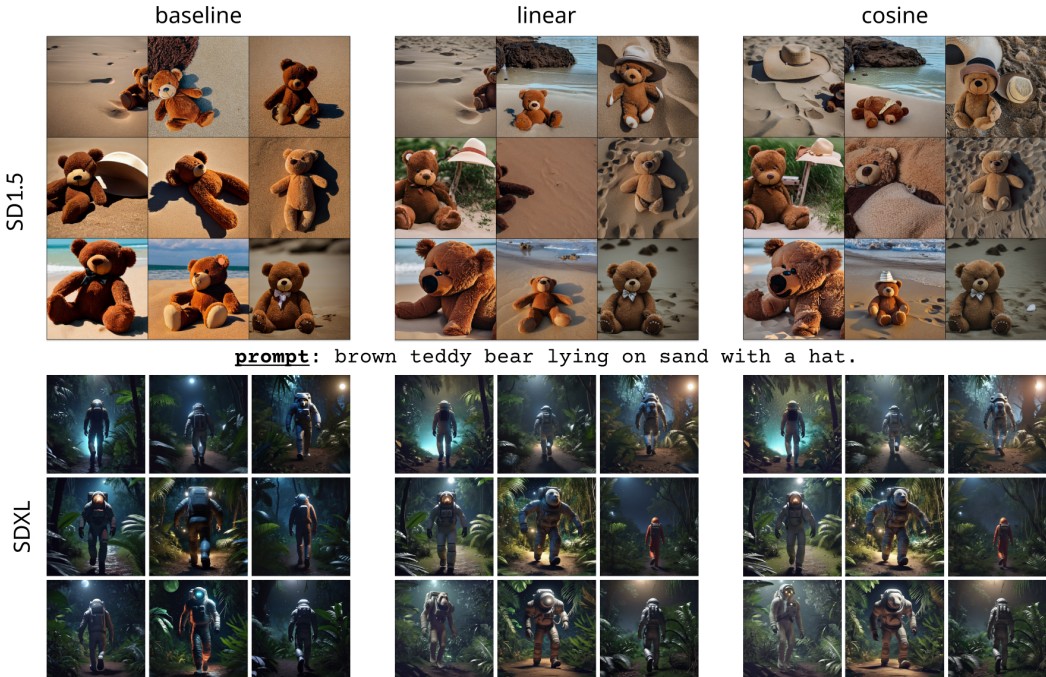

Figure 4: **Qualitative results of diversity** of different guidance schedulers compared with static baseline. Heuristic schedulers show better diversity: more composition and richer background types for the teddy bear example, as well as the gesture, lighting, colour and compositions in the astronaut image.

However, this requires training a noise-dependent classifier externally, which can be cumbersome and impractical for novel classes or more complex conditions e.g. textual prompts. For this reason, with an implicit classifier from Bayes rule $\nabla_{x_t} \log p(c|x_t) = \nabla_{x_t} \log p(x_t, c) - \nabla_{x_t} \log p(x_t)$, Ho & Salimans (2021) propose to train a diffusion network on the joint distribution of data and condition by replacing $\epsilon_\theta(x_t)$ with $\epsilon_\theta(x_t, c)$ in $L_{\text{simple}}$. By dropping the condition during training, they employ a single network for both $\nabla_{x_t} \log p(x_t, c)$ and $\nabla_{x_t} \log p(x_t)$. This gives the *Classifier-Free Guidance (CFG)*, also controlled by $\omega$:

$$\hat{\epsilon}_\theta(x_t, c) = \epsilon_\theta(x_t, c) + \omega \left( \epsilon_\theta(x_t, c) - \epsilon_\theta(x_t) \right) \quad . \tag{3}$$

We can reformulate the above two equations into two terms: a *generation* term $\epsilon_\theta(x_t) \propto \nabla_{x_t} \log p(x_t)$ and a *guidance* term $\nabla_{x_t} \log p(c|x_t)$. The guidance term can be derived either from a pre-trained classifier or an implicit one, with $\omega$ balancing between generation and guidance.

## 4 Towards dynamic guidance: Should guidance be constant?

Our initial experiments show that removing guidance at certain timesteps can improve performance. This is in line with concurrent work (Kynkäänniemi et al., 2024). To further investigate this, we conducted a negative perturbation analysis experiment to determine the impact of the guidance across all timesteps.

**Negative Perturbation Analysis.** This analysis is on the CIFAR-10 dataset: a 60,000 images dataset with a $32 \times 32$ resolution, distributed across 10 classes. We choose the original DDPM method (Ho et al., 2020) denoising on pixel space as the backbone and class-conditioning guidance.

To investigate the importance of guidance across different timestep intervals, we first employ static guidance of $\omega = 1.15$, then independently set the guidance to zero across different 50-timestep intervals (20 intervals in total spanning all timesteps), and compute the FID for each of these piece-wise zeroed guidance schedulers of 50,000 generated images. If we mathematically model the removal method, it can approximate a family

of parameterized gate/inverse gate functions with two parameters defining the starting point $s$ and size of the kernel $d$: $g(t) = 1 - (H(t - s) - H(t - (s + d)))$, where $H$ is Heaviside step function.

The results are illustrated in Figure 6b. For example, the second data point on the left of the curve represents the FID performance when guidance is removed only in the interval $t = [50, 100]$ while maintaining static at others. We observe multiple phenomena from the results: (1) non-constant guidance curve can outperform static guidance in terms of FID; (2) zeroing the guidance at earlier stages improves the FID performance; (3) zeroing the guidance at later stages significantly degrades it.

However, this removal scheme is not practical for real usage: (i) grid searching two parameters requires generating a prohibitively costly number of images; (ii) as shown in Section 6, parameterized methods often fail to generalize; (iii) further investigation, detailed in Appendix Section B, demonstrates that instead of completely removing CFG from some timesteps, keeping it with lower values increases the performance.

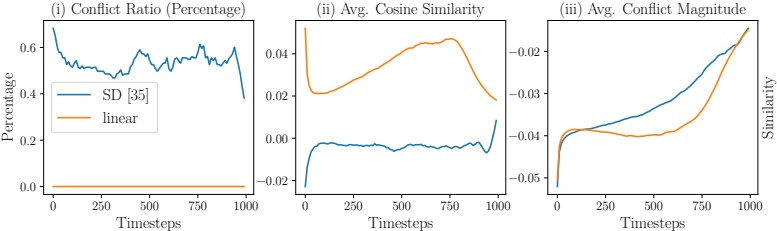

Figure 5: **Visualization of Conflicted Terms** from SD1.5 Rombach et al. (2022) shows that static guidance presents conflicts, while a guidance scheduler reduces the conflict between generation and guidance terms.

**Conflicted terms.** Our assumption is that the *guidance* and *generation* terms (see Eq. 3) may be adversarial during inference. Following (Dinh et al., 2023a), Figure 5 quantifies the conflict by measuring (a) the ratio of negative cosine similarity; (b) average cosine similarity (i.e. directional conflict, -1 and 1 for maximum and minimum conflicts); and (c) conflict magnitude (Dinh et al., 2023a), defined as: $\Phi(\epsilon_1, \epsilon_2) = \frac{-2|\epsilon_1|_2|\epsilon_2|_2}{|\epsilon_1|_2^2 + |\epsilon_2|_2^2}$ where $\epsilon$ is each term at each timestep, with $\Phi$ resulting $-1$ and $0$ indicates zero and maximum conflict. We evaluate 1000 generation from COCO prompts (Lin et al., 2014) with SD1.5 and show in Figure 5(i) that SD (orange) exhibits $\sim 50\%$ conflict ratio along the generation with high magnitude conflict (see Figure 5(iii)(right) with curves closer to zero). When the guidance is lowered at the beginning (e.g. linearly increasing as shown in Section 5), less conflict for both magnitude and directional metrics is shown in all subfigures blue curves.

**Dynamic guidance.** Having observed that removing the guidance at certain timesteps improves the performance over using a *static* weight $\omega$ for CFG like in Ho & Salimans (2021); Dhariwal & Nichol (2021) and reducing the guidance at beginning linearly can reduce the conflict, we ask the question of whether we can replace static guidance with other options. Therefore, we investigate *dynamic* guidance scheduler that evolves throughout the generation process, which is also in line with some empirical schemes mentioned in recent literature (Blattmann et al., 2023; Chang et al., 2023; Donahue et al., 2018). In that case, the CFG Equation 3 is rewritten as follows:

$$\hat{\epsilon}_\theta(x_t, c) = \epsilon_\theta(x_t, c) + \omega(t)\left(\epsilon_\theta(x_t, c) - \epsilon_\theta(x_t)\right) \quad . \tag{4}$$

To identify an effective dynamic scheduler $\omega(t)$, we analyse two types of function in subsequent sections: parameter-free heuristic schedulers in Section 5 and single-parameter parameterized ones in Section 6.

## 5 Dynamic Guidance: Heuristic Schedulers

We first use six simple heuristic schedulers as dynamic guidance $\omega(t)$, split into three groups depending on the shape of their curve: (a) increasing functions (linear, cosine); (a) decreasing functions (inverse linear,

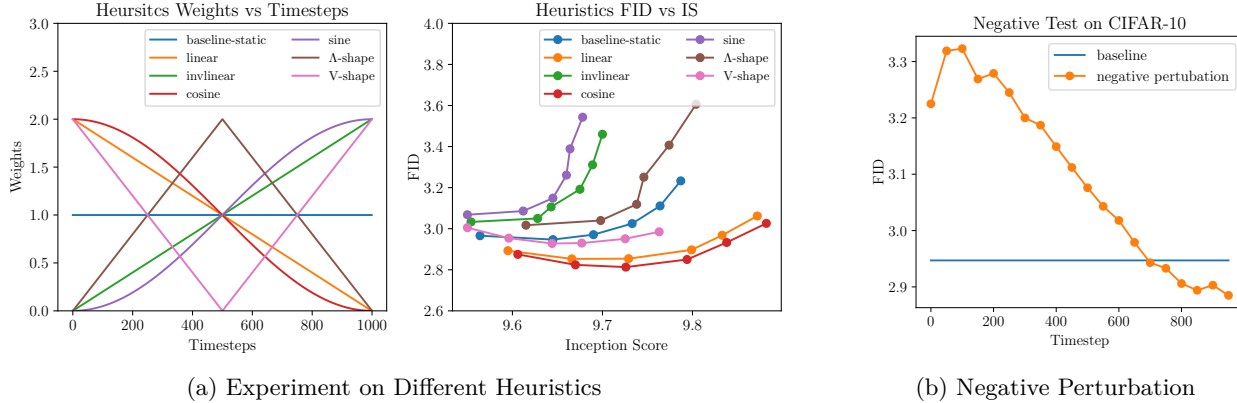

(a) Experiment on Different Heuristics

(b) Negative Perturbation

Figure 6: **Preliminary Analysis on CIFAR-10 (a) Various heuristic curves** with their FID vs. IS performances. **(b) Negative perturbation** by setting the guidance scale to 0 across distinct intervals while preserving static guidance to the rest. By eliminating the weight at the **initial stage** (e.g., $T = 800$), the lowered FID shows an enhancement, whereas removing guidance at higher timesteps leads to worse FID.

sine); (c) non-monotonic functions (linear V-shape, linear $\Lambda$-shape), defined as:

$$\text{linear: } \omega(t) = 1 - t/T,$$
$$\text{cosine: } \omega(t) = \cos{(\pi t/T)} + 1,$$
$$\text{V-shape: } \omega(t) = \text{invlinear}(t) \; if \; t < T/2,$$
$$\text{linear}(t) \; else,$$

$$\text{invlinear: } \omega(t) = t/T,$$
$$\text{sine: } \omega(t) = \sin{(\pi t/T - \pi/2)} + 1,$$
$$\Lambda\text{-shape: } \omega(t) = \; \text{linear}(t) \; if \; t < T/2,$$
$$\text{invlinear}(t) \; else.$$

To allow for a direct comparison between the effect of these schedulers and the static guidance $\omega$, we normalize each scheduler by the area under the curve. This ensures that the same *amount of total guidance* is applied over the entire denoising process, and allows users to rescale the scheduler to obtain a behavior similar to that of increasing $\omega$ in static guidance. More formally, this corresponds to the following constraint: $\int_0^T \omega(t)dt = \omega T$. For example, this normalization leads to the corresponding normalized linear scheduler $\omega(t) = 2(1 - t/T)\omega$. We show in Figure 6a (left) the different normalized curves of the 6 schedulers.

## 5.1 Class-conditional image generation with heuristic schedulers

**Heuristic Schedulers Analysis.** We first study the 6 previously defined heuristic schedulers $\omega(t)$ on the CIFAR-10-DDPM setting for class-conditional synthesis same as in the Section 4. To assess the performance, we use the Frechet Inception Distance (FID) and Inception Score (IS) metrics, over $50,000$ inference from 50-step DDIM (Song et al., 2020). In this experiment, we evaluate the impact of a range of different guidance total weight: $[1.1, 1.15, 1.2, 1.25, 1.3, 1.35]$, to study its influence over the image quality vs class adherence trade-off. We show the results in Figure 6a, right panel and observe that both increasing schedulers (linear and cosine) significantly improve over the static baseline, whereas decreasing schedulers (invlinear and sine) are significantly worse than the static. The V-shape and $\Lambda$-shape schedulers perform respectively better and worse than the static baseline, but only marginally.

**Preliminary Conclusion.** Combining with the observation from Section 4 that removing the beginning stage improves the performance, they point to the same conclusion: **monotonically increasing guidance schedulers** achieve improved performances, revealing that the static CFG primarily may overshoot the guidance in the initial stages. In the remainder of this work, we only consider monotonically increasing schedulers, as we consider these findings sufficient to avoid examining all other schedulers on other tasks. (more details in Appx. and Figure 2 shows a qualitative results in SDXL)

**Experiments on ImageNet.** On ImageNet, we explore the linear and cosine schedulers that performed best on CIFAR-10. In Figure 7d, we observe that the linear and cosine schedulers lead to a significant

improvement over the baseline, especially at higher guidance weights, enabling a better FID/Inception Score trade-off. More experiments in Appx. lead to a similar conclusion.

### 5.2 Text-to-image generation with heuristic schedulers

We study the linear and the cosine scheduler on text-to-image generation. The results for all proposed heuristics are in Appx. Tables 12 and 14, where we observe a similar trend as before: heuristic functions with increasing shape report the largest gains on both SD1.5 and SDXL.

**Dataset and Metrics.** We use text-to-image models pre-trained on LAION (Schuhmann et al., 2022), which contains 5B high-quality images with paired textual descriptions. For evaluation, we use the COCO (Lin et al., 2014) val set with $30,000$ text-image paired data.

We use three metrics: (i) *Fréchet inception distance (FID)* for the fidelity of generated images; (ii) *CLIP-Score (CS)* (Radford et al., 2021) to assess the alignments between the images and their corresponding text prompts; (iii) *Diversity (Div)* to measure the model's capacity to yield varied content. For this, we compute the standard deviation of image embeddings via Dino-v2 (Oquab et al., 2023) from multiple generations of the same prompt (more details for Diversity in Appendix).
We compute FID and CS for a sample set of $10,000$ images against the COCO dataset in a zero-shot fashion (Rombach et al., 2022; Saharia et al., 2022). For diversity, we resort to two text description subsets from COCO: 1000 *longest captions* and *shortest captions* each (-L and -S in Figure 7a) to represent varying descriptiveness levels; longer captions provide more specific conditions than shorter ones, presumably leading to less diversity. We produce 10 images for each prompt using varied sampling noise.

**Model.** We experiment with two models: (1) Stable Diffusion (SD) (Rombach et al., 2022), which uses the CLIP (Radford et al., 2021) text encoder to transform text inputs to embeddings. We use the public checkpoint of SD v1.5 [1] and employ DDIM sampler with *50* steps. (2) SDXL (Podell et al., 2023), which is a larger, advanced version of SD (Rombach et al., 2022), generating images with resolutions up to *1024* pixels. It leverages LDM (Dhariwal & Nichol, 2021) with larger U-Net architectures, an additional text-encoder (OpenCLIP ViT-bigG), and other conditioning enhancements. We use the SDXL-base-1.0[2] (SDXL) version without refiner, sampling with DPM-Solver++ (Lu et al., 2022b) of *25 steps*.

**Results.** We show the FID vs. CS curves in Figure 7a, 7c for SD and SDXL respectively (more tables in Appx. Section G). We expect an optimal balance of a high CS and a low FID (i.e., the right-down corner).

**Analysis on SD (Figure 7a)**. For FID vs CS, the baseline (Rombach et al., 2022) yields inferior results compared to the linear and cosine heuristics with linear recording lower FID. The baseline regresses FID fast when CS is high, but generates the best FID when CS is low, i.e., low condition level. This, however, is usually not used for real applications, e.g., the recommended $\omega$ value is 7.5 for SD1.5, highlighted by the dotted line in Figure 7a with the black solid arrow representing the gain of heuristic schedulers on FID and CS respectively. For Div vs CS, heuristic schedulers outperform the baseline (Rombach et al., 2022) on both short (S) and long (L) captions at different guidance scales. Also, cosine shows superiority across the majority of the CS range. Overall, heuristic schedulers achieve improved performances in FID and Diversity, recording 2.71(17%) gain on FID and 0.004(16%) gain (of max CS-min CS of baseline) on CS over $\omega$=7.5 default guidance in SD. Note, this gain is achieved *without* hyperparameter tuning or retraining.

**Analysis on SDXL (Figure 7c).** In FID, both the linear and cosine schedulers achieve better FID-CS than the baseline (Podell et al., 2023). In Diversity, linear is slightly lower than cosine, and they are both better than static baseline. Additionally, unlike the baseline (blue curves) where higher guidance typically results in compromised FID, heuristic schedulers counter this.

**User study.** We present users with a pair of mosaics of 9 generated images and ask them to vote for the best in terms of realism, diversity and text-image alignment. Each pair compares static baseline generations against cosine and linear schedulers. Figure 7b reports the results. We observe that over 60% of users consider scheduler-generated images more realistic and better aligned with the text prompt, while approximately

---

[1] https://huggingface.co/runwayml/stable-diffusion-v1-5
[2] https://github.com/Stability-AI/generative-models

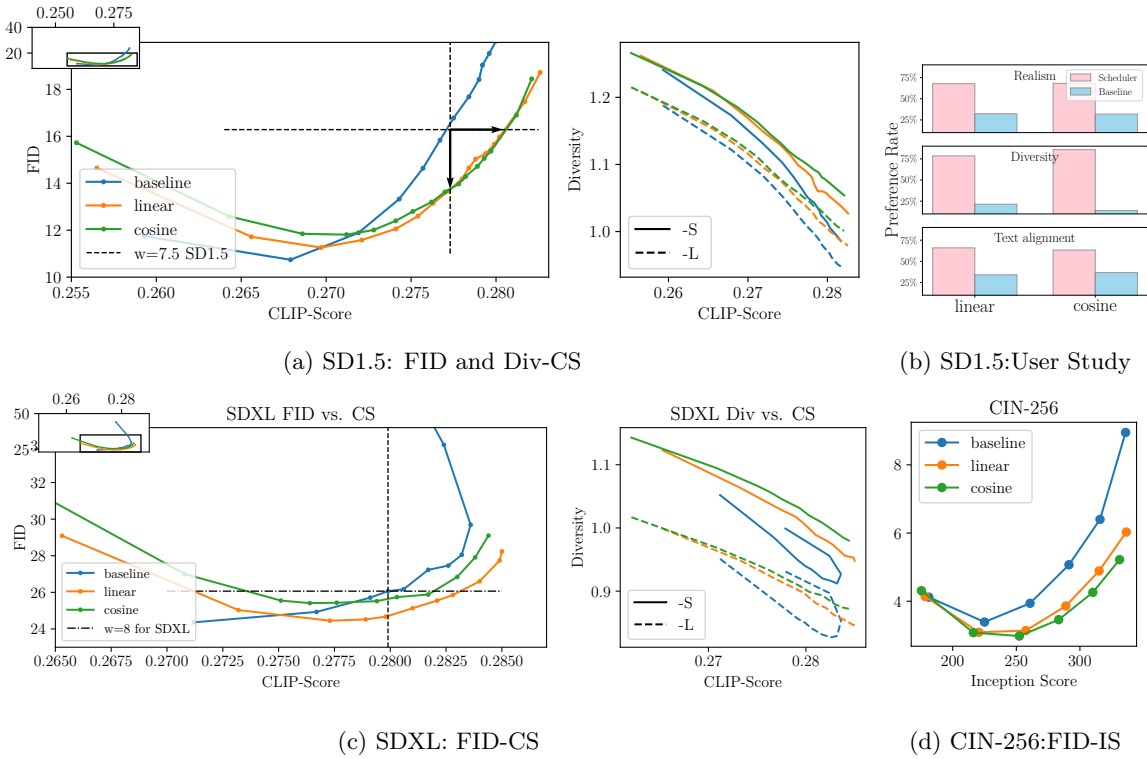

Figure 7: **Class-conditioned and text-to-image generation results of monotonically-increasing heuristic schedulers (linear and cosine). (a) FID and Div vs. CS** for SD1.5 (Rombach et al., 2022). We highlight the gain of FID and CS compared with the default $\omega$=7.5 with black arrows, diversity on the right shows that the heuristic guidance performs better than static baseline guidance; **(b) our user study** also reveals that images generated with schedulers are consistently preferred than the baseline in realism, diversity and text alignment; **(c) results for SDXL (Podell et al., 2023) on FID and Div vs. CS** with similar setup to (a); **(d) CIN-256 LDM** (Dhariwal & Nichol, 2021) are assessed with FID vs. IS. Heuristic schedulers outperform the baseline static guidance on fidelity and diversity across multiple models.

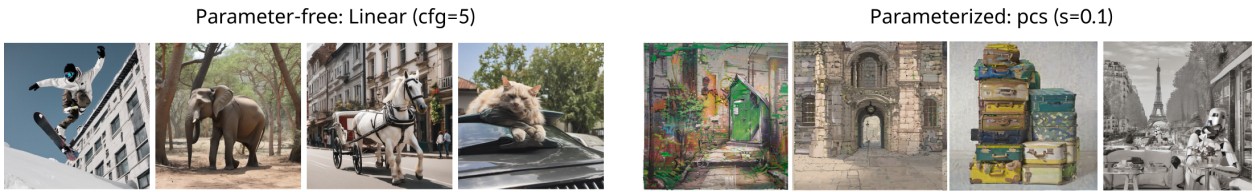

Figure 8: **Failure cases** of parameter-free and parameterized approaches: monotonically increasing guidance may mute the guidance at the beginning (especially when overall guidance is low), causing structural errors; and incorrectly chosen parameters can lead to fuzzy details and low saturation problems.

80% find guidance schedulers results more diverse. This corroborates our hypothesis that static weighting is perceptually inferior to dynamic weighting. More details in Appx.

**Qualitative results.** Figure 3 depicts the fidelity of various sets of text-to-image generations from SD and SDXL. Heuristic schedulers (linear and cosine) enhance the image fidelity: better details in petals and leaves of the flower images, as well as the texture of bird features. In the arches example, we observe more natural colour shading as well as more detailed figurines with reflective effects. Figure 4 showcases the diversity of outputs in terms of composition, color palette, art style and image quality by refining shades and enriching textures. Notably, the teddy bear shows various compositions and better-coloured results than the baseline,

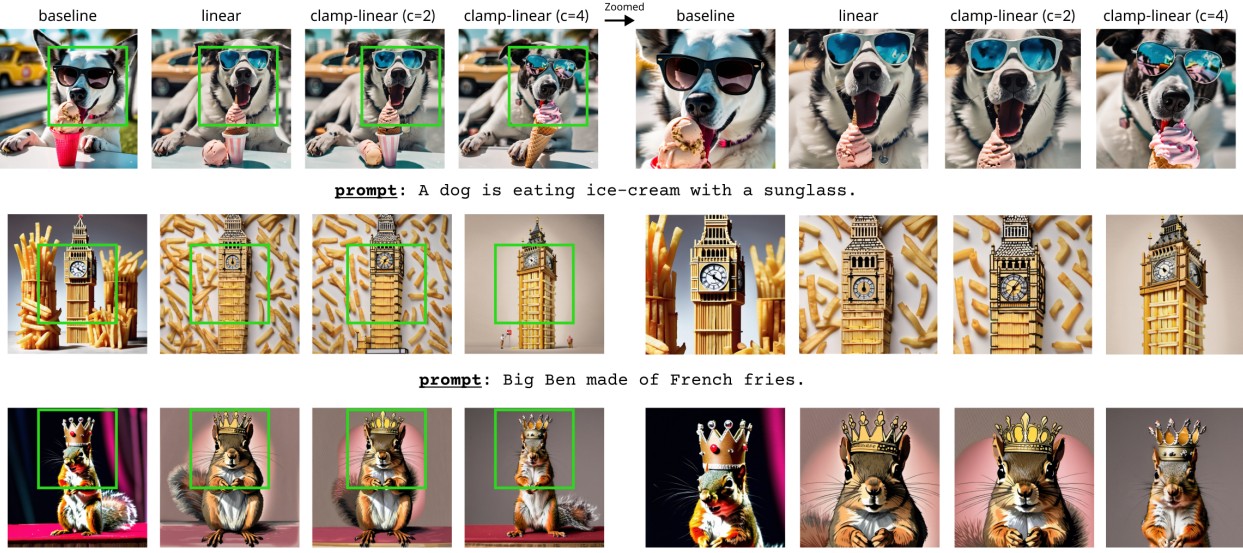

Figure 9: **Qualitative comparison** of baseline, linear and clamp-linear on SDXL. Both dynamic schedulers are better than the baseline, clamp-linear with $c=4$ outperforms all with better details and higher fidelity.

which collapsed into similar compositions. Similarly, in the astronaut example, the baseline generates similar images while heuristic schedulers reach more diverse character gestures, lighting and compositions.

### 5.3 Findings with heuristic schedulers

In summary, we make the following observations: monotonically increasing heuristic schedulers (e.g., linear and cosine) (a) improve generation performances (IS/CS vs. FID) over static baseline on different models; (b) improve image fidelity (texture, details), diversity (composition, style) and quality (lighting, gestures). We note that this gain is achieved without hyperparameter tuning, retraining or extra computational cost.

**Failure cases.** For the failure cases involving monotonically increasing guidance, we observe that *undershooting* the guidance scale during the initial stages can compromise the structural integrity of the generated outputs. This often results in anatomical and geometric errors, such as the appearance of a third leg in humans, a fifth leg in quadruped animals, or incorrect spatial relationships, as illustrated in Figure 8.

## 6 Dynamic Guidance: Parametrized Schedulers

We investigate two parameterized schedulers with a tunable parameter to maximize performance: a power-cosine curve family (introduced in MDT (Gao et al., 2023)) and two clamping families (linear and cosine).

The parameterized family of powered-cosine curves (**pcs**) and clamping (**clamp**) is defined by the controllable parameter $s$ and $c$ respectively:

$$w_t = \frac{1 - \cos \pi \left( \frac{T-t}{T} \right)^s}{2} w \qquad \textbf{(pcs)} \qquad (5)$$

$$w_t = \max(c, w_t) \qquad \textbf{(clamp)} \qquad (6)$$

In our work, we use clamp-linear but this family can be extended to other schedulers (more in sup. mat.). Our motivation lies in our observation that excessive muting of guidance weights at the initial stages can compromise the structural integrity of prominent features. This contributes to bad FID at lower values of $\omega$ in Figure 7a, suggesting a trade-off between model guidance and image quality. However, reducing guidance

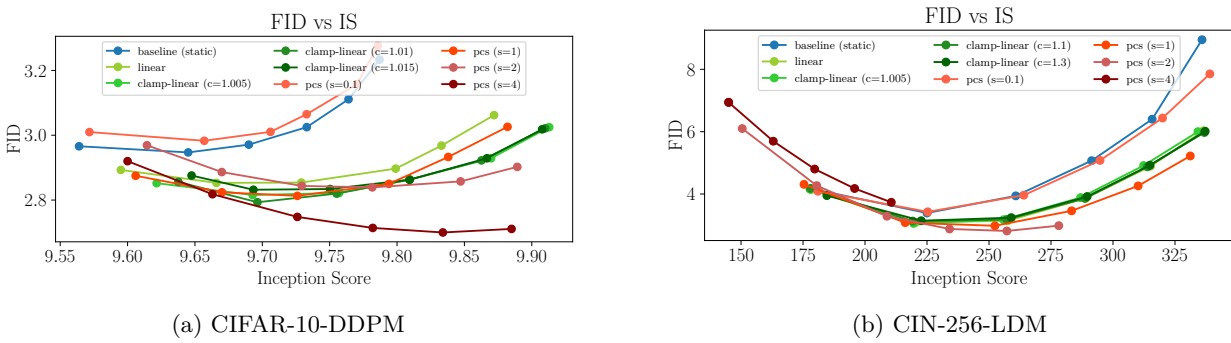

(a) CIFAR-10-DDPM

(b) CIN-256-LDM

Figure 10: **Class-conditioned generation results** of parameterized clamp-linear and pcs on (a) CIFAR-10-DDPM and (b) CIN-256-LDM. Optimising parameters improves performances but these parameters do not generalize across models and datasets.

intensity early in the diffusion process is also the origin of enhanced performances, as shown in Section 5. This family represents a trade-off between diversity and fidelity while giving users precise control.

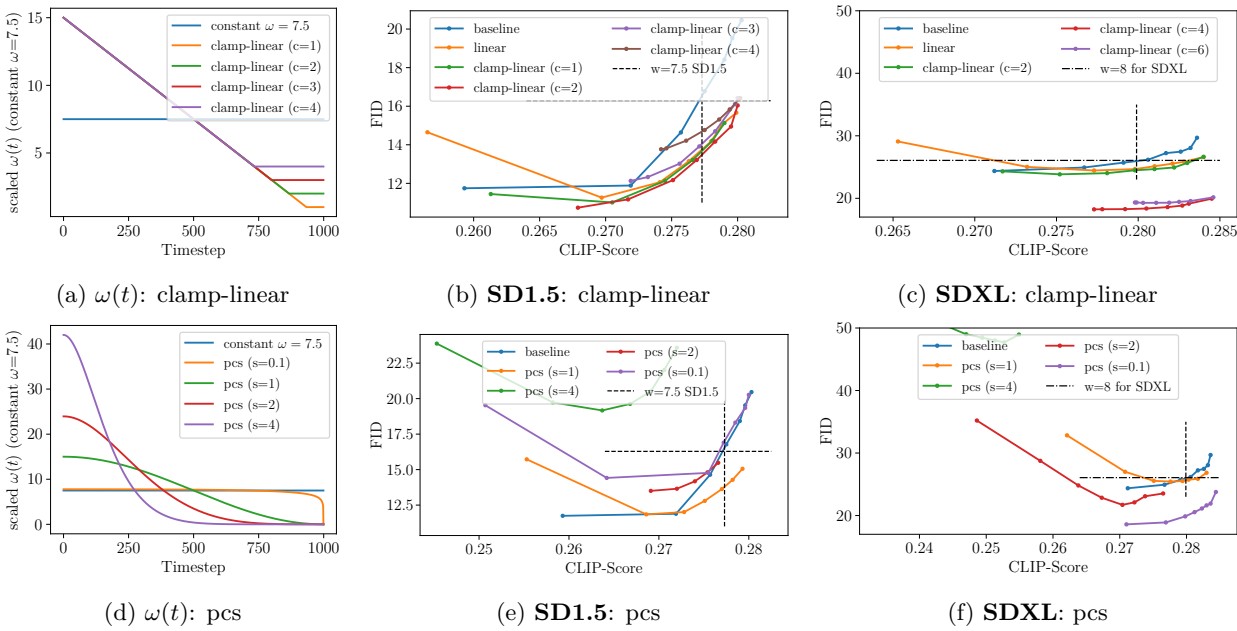

(a) $\omega(t)$: clamp-linear

(b) **SD1.5**: clamp-linear

(c) **SDXL**: clamp-linear

(d) $\omega(t)$: pcs

(e) **SD1.5**: pcs

(f) **SDXL**: pcs

Figure 11: **Text-to-image performance for two parameterized schedulers: clamp-linear and pcs**. For clamp-linear, **(a)** shows the guidance curves for different parameters and **(b,c)** displays the FID vs. CS for SD1.5 and SDXL, respectively. For pcs, **(d)** shows the guidance curves and **(e,f)** depicts the FID vs. CS. Optimal parameters for either clamp or pcs outperform the static baseline for both SD1.5 and SDXL.

## 6.1 Class-conditional image generation with parametrized schedulers

We experiment with two parameterized schedulers: clamp-linear and pcs on CIFAR10-DDPM (Figure 10a) and ImageNet(CIN)256-LDM (Figure 10b). We observe that, for both families, tuning parameters improves results over baseline and heuristic schedulers. The optimal parameters are $c=1.01$ for clamp-linear and $s=4$ for pcs on CIFAR10-DDPM, vs $c=1.1$ for clamp-linear and $s=2$ for pcs on CIN-256. Overall, parameterized schedulers improve performances; however, the optimal parameters do not apply across datasets and models.

## 6.2 Text-to-image generation with parametrized schedulers

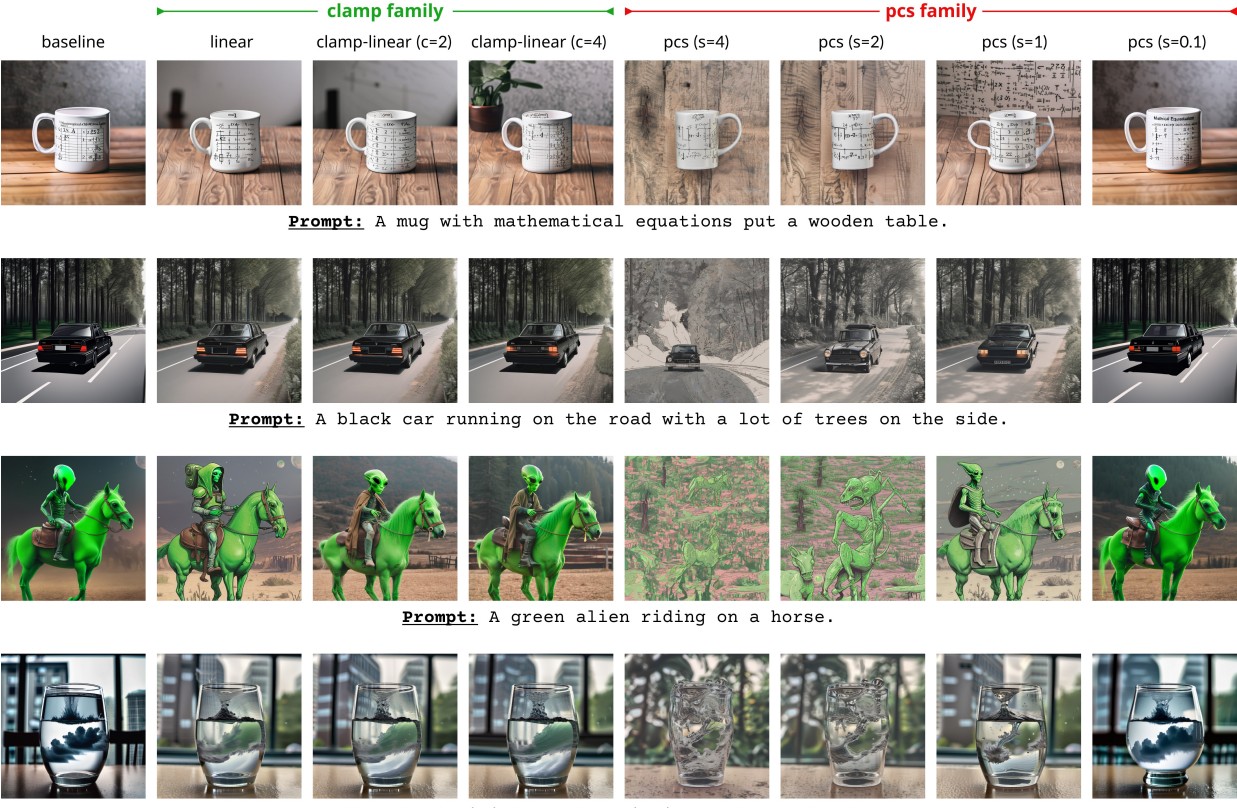

Figure 12: **Qualitative results** for parametrized schedulers clamp-linear and pcs on SDXL Podell et al. (2023). Overall, $c$=4 for clamp-linear gives the most visually pleasing results.

We experiment with two parameterized schedulers: clamp-linear (clamp-cosine in sup. mat.) and pcs, with their guidance curves in Figures 11a,11d, respectively.

For SD1.5 (Rombach et al., 2022), the FID vs. CS results are depicted in Figures 11b and 11e. The pcs family struggles to achieve low FID, except when $s = 1$. Conversely, the clamp family exhibits optimal performance around $c$=2, achieving the best FID and CLIP-score balance while outperforming all pcs values.

For SDXL (Podell et al., 2023), the FID vs. CS results are depicted in Figures 11c and 11f. The pcs shows the best performance at $s = 0.1$. Clamp-linear achieves optimum at $c = 4$ (FID 18.2), largely improving FID across the entire CS range compared to the baseline (FID 24.9, about 30% gain) and the linear scheduler.

The optimal parameters of clamp-linear (resp. pcs) are not the same for both models, i.e. $c$=2 for SD1.5 and $c$=4 for SDXL (resp. $s$=1 and $s$=0.1 for pcs). This reveals the lack of generalizability of this family.

**Qualitative results.** The results of Figure 9 further underscore the significance of choosing the right clamping parameter. This choice markedly enhances generation performance, as evidenced by improved fidelity (dog and squirrel image), textual comprehension (Fries Big Ben), and details (sunglasses).

Figure 12 compares two parameterized families: (i) clamp and pcs (Gao et al., 2023), where the clamp performs best at $c = 4$ and the pcs at $s = 1$. We observe that the clamp-linear $c = 4$ demonstrates better details (e.g., mug, alien), realism (e.g., car, storm), and more textured backgrounds (e.g., mug, car). Although $s = 4$ for pcs leads to the best results for class-conditioned generation, we see that the pcs in text-to-image task tends to over-simplify content, produce fuzzy images (e.g., mug) and deconstructed composition. This highlights our argument that optimal parameters do not generalize across datasets or tasks.

**Rectified flow model.** Recent advancements in Rectified Flow (RF) (Liu et al., 2022) improve generative models with straighter and shorter generation trajectories. In this section, we show the performance of guidance schedulers on RF-based methods e.g. Stable Diffusion 3 (SD3). Our experiments involved calculating the FID and CS on the COCO dataset, similar to all previous experiments. The results, shown in Figure 13a, reveal three key findings: (1) dynamically increasing schedulers can further enhance the generation capabilities of RF methods, e.g. SD3 ($\sim$ 6 FID (gain more than $\sim$ 20%) at the same CS); (2) applying clamping leads to additional improvements; and (3) the optimal hyperparameters ($c = 0.5$) does **not generalize** to other models (SD1.5, SDXl), which is in line with our central hypothesis. The qualitative results presented in Figure 13b and in appendix Figure 24 also demonstrate that the guidance scheduler and clamping methods enhance the details (castle, ship), tone (night and sea), and composition (fishes) of the generation.

**Text-to-text Image Translation** To show the generalization of schedulers other than text- or label-conditioned tasks, we experiment with image-to-image translation on SD1.5 (Rombach et al., 2022), with details in Appendix A. Results show improved FID-CS balance for linear schedulers, with no clamping being optimal. This differs from the SD1.5 T2I task, showing that the optimal parameter does not generalize.

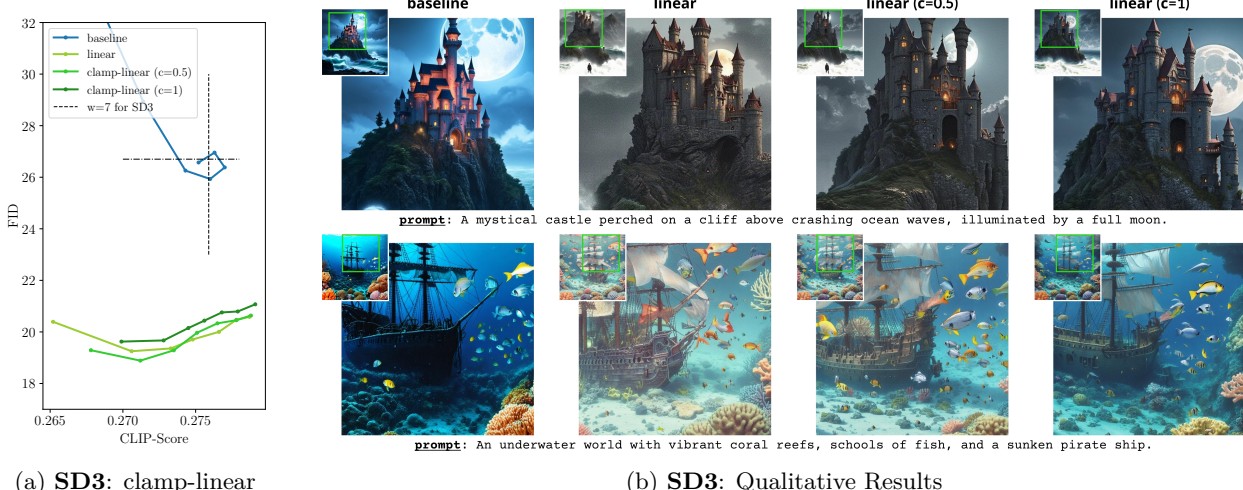

(a) **SD3**: clamp-linear

(b) **SD3**: Qualitative Results

Figure 13: **Text-to-image performance and qualitative results** on SD3. We show that (a) both linear and clamp-linear guidance schedulers enhance the balance between FID and CLIP score (CS), and (b) the generated images exhibit improved detail and higher fidelity.

### 6.3 Findings with parametrized schedulers

Our observations are: (a) tuning the parametrized functions improves the performance for both generation tasks, (b) tuning clamps seems easier than pcs family, as its performance shows fewer variations, and (c) the optimal parameters for one method does not generalize across different settings. Thus, specialized tuning is required for each model and task, leading to extensive grid searches and increased computational load.

**Failure cases.** The main risk with parameterized functions arises from **ill-chosen parameters**. As shown in Figures 8, 12 20, poorly chosen parameters, particularly for pcs family, often lead to overshooting at the later stages, resulting in fuzzy detail, broken structure and unnatural colour in the generated images.

## 7 Conclusion

We analyzed dynamic schedulers for the weight parameter in Classifier-Free Guidance by systematically comparing heuristic and parameterized schedulers. We experiment on two tasks (class-conditioned generation and text-to-image generation), several models (DDPM, SD1.5 and SDXL) and various datasets.

**Discussion.** Our findings are: (1) a simple monotonically increasing scheduler systematically improves the performance compared to a constant static guidance, at no extra computational cost and with no hyper-parameter search. (2) parameterized schedulers with tuned parameters per task, model and dataset, improve the results. They, however, do not generalize well to other models and datasets as there is no universal parameter that suits all tasks.

For practitioners who target state-of-the-art performances, we recommend searching or optimizing for the best clamping parameter. For those not willing to manually tune parameters per case, we suggest using heuristics, specifically linear or cosine.

## 8 Acknowledgment

This work was supported by ANR APATE ANR-22-CE39-0016, ANR TOSAI ANR-20-IADJ-0009, Hi!Paris grant and fellowship, and by the European Union's Horizon 2020 research and innovation programme under the Marie Curie grant agreement No 860768 (CLIPE project). It was granted access to the High-Performance Computing (HPC) resources of IDRIS under the allocations 2024-AD011014300R1 made by GENCI.

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

## Appendix

In this appendix, we provide additional content covering: (A) an extra experiment about the task of image-to-image translation; (B) an ablation study to demonstrate the necessity of CFG at all time intervals; (C) a toy example to explain the mechanism and rationale of the dynamic weighted scheduler; (D) an additional comparison of parameterized function-based dynamic schedulers; (E) more qualitative results; (F) ablation experiments on different aspects of dynamic weighting schedulers; (G) a list of tables of all results demonstrated; (H) detailed design of user study. Following is the table of contents

## A    Image-to-image Translation Task

In addition to the image generation task, we also evaluated the image-to-image translation task to demonstrate the generalization capabilities of the dynamic guidance scheduler across multiple conditioning scenarios. The experimental setup closely follows the one outlined in the main manuscript (Section 6), using the SD1.5 backbone model (Rombach et al., 2022) and images and their correspondent prompts from COCO dataset (Lin et al., 2014) to achieve image-to-image translation task. As shown in Figure 14a, the linear dynamic guidance scheduler significantly improves the FID vs. CLIP-Score trade-off ($\sim 2$ FID at the same CLIP-Score) in image-to-image translation tasks. However, the optimal parameter for the parameterized scheduler was found to involve clamping at guidance level $c = 0$, which differs from the optimal parameters identified in the generation task ($c = 2$). This further supports our primary claim that a generalized parameterized scheduler does not exist across different tasks. The qualitative results in Figure 14b also showed better structure (car), details (bird background) and prompt understanding (graffiti in the bin).

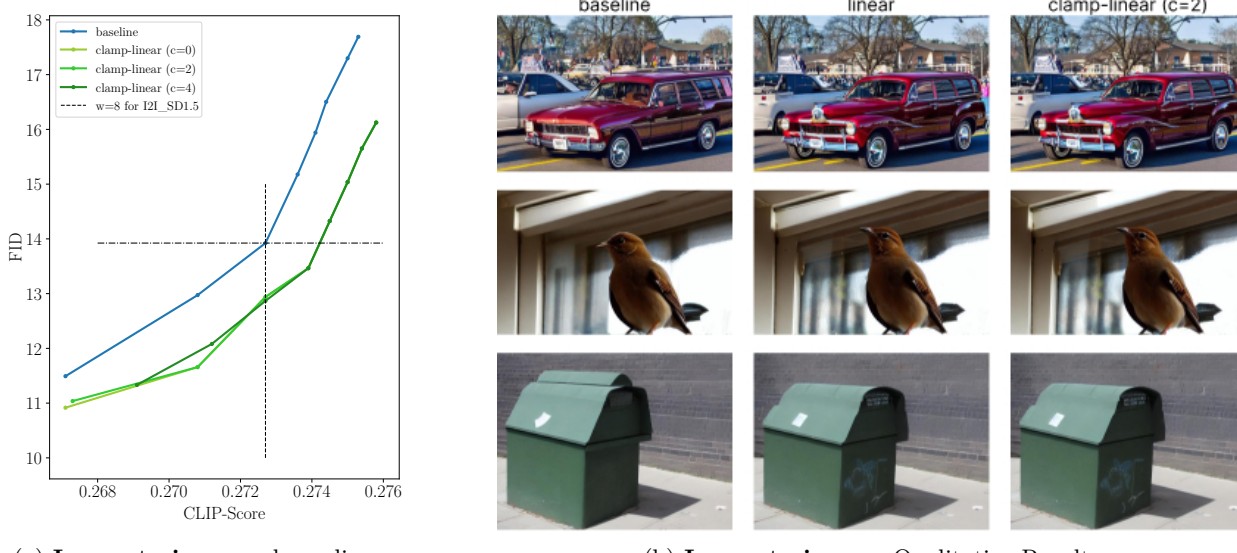

(a) **Image-to-image**: clamp-linear

(b) **Image-to-image**: Qualitative Results

Figure 14: **Image-to-image performance and qualitative results** on SD1.5. We show that (a) both linear and clamp-linear guidance schedulers enhance the balance between FID and CLIP score (CS) of the image-to-image translation task, and (b) the generated images exhibit improved detail and higher fidelity.

## B    The necessity of CFG at all time interval

Recent concurrent work Kynkäänniemi et al. (2024); Zhang et al. (2024); Castillo et al. (2023) has suggested that partially removing the classifier-free guidance (CFG) (beginning or ending) could enhance generation performance or achieve the acceleration with minimal performance influence. For instance, Kynkäänniemi et al. (2024) proposes removing the initial and final timesteps of CFG, retaining only a middle interval for the guidance process. In this section, we conduct two ablation studies on SD1.5 to confirm that the CFG can be required across all time intervals.

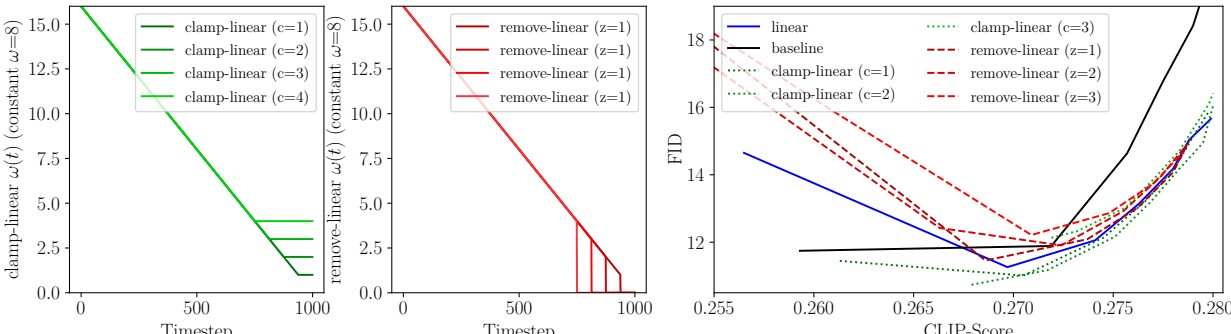

Figure 15: **Comparison between guidance removal (labelled as z) and clamping (labelled as c) on SD1.5**. We see that CFG removal scheme shows improvement compared to the static guidance baseline (black) but is worse than the linear guidance scheduler (blue solid) and clamping schemes (red dotted).

**The necessity of CFG at the beginning stage** As demonstrated in Figure 6b, we explore the impact of negative perturbation and ablation of all heuristic functions in Figure 12. Generally, employing a lower guidance level in the initial stages can enhance performance compared to static guidance. To analyse the effectiveness of guidance removal (setting to zero), static guidance, the linear scheduler, and the clamping scheme, we conducted experiments on SD1.5. Guidance is removed at the same timestep as the clamping transition point; rather than clamping guidance to a hyperparameter constant, we reduce it completely to zero (Figure 15 left two panels). The results, depicted in Figure 15 right panel, show that while guidance removal at the beginning stage (red dotted curve) indeed improves performance compared to the static baseline (black solid curve), both the linear scheduler (blue solid curve) and clamping schemes (green dotted lines) achieve better balances of FID vs. CLIP-Score (CS).

**The necessity of CFG at the ending stage** Zhang et al. (2024); Castillo et al. (2023) suggest that removing the final stage of guidance could accelerate generation by directly replacing the CFG with conditional or unconditional outputs. However, as shown in Figure 6b(b), our analysis indicates that removing this stage can reduce performance for specific tasks. Despite this, the possibility of safely removing the ending stage guidance does not contradict our argument that **enhancing the end could improve performance**. To further confirm this, we conducted an ablation experiment comparing the effects of removing versus boosting the final guidance intervals. In this experiment, 10%, 20%, and 30% of the ending guidance were either removed or increased by a factor of 1.5. The results, presented in Table 1, reveal: (i) removing or boosting the ending guidance has a marginal impact on the CLIP-Score; (ii) elimination of guidance can lead to a regression in performance; and (iii) boosting the guidance can significantly enhance FID, with gains of 0.54 and 0.8 in FID when boosting the final 30% of guidance by 1.5×.

In conclusion, based on the results from two previous ablation studies, we confirm that **an adequate level of guidance is necessary at all intervals** of the generation process. While removing parts of the guidance can accelerate the process, it results in underperformance when compared to our analyzed heuristic monotonically increasing guidance scheduler, such as linear, and also when compared to well-tuned parameterized functions, such as the clamping method.

## C   A toy example of fidelity vs condition adherence

Knowing the equation of CFG can be written as a combination between a *generation term* and a *guidance term*, with the second term controlled by guidance weight $\omega$:

$$\hat{\epsilon}_\theta(x_t, c) = \epsilon_\theta(x_t, c) + \omega\left(\epsilon_\theta(x_t, c) - \epsilon_\theta(x_t)\right) \quad . \tag{7}$$

To better understand the problems in diffusion guidance, we present a toy example, where we first train a diffusion model on a synthetic dataset of $50,000$ images ($32 \times 32$) from two distinct Gaussian distributions:

Table 1: Impact of removing/boosting CFG at the end with SD1.5.

| Guidance scale | $\omega$=8 | | $\omega$=11 | |
|---|---|---|---|---|
| Method | Clip-Score | FID | Clip-Score | FID |
| static | 0.2775 | 16.78 | 0.2792 | 19.03 |
| remove final 10% | 0.2777 | 17.61 | 0.2792 | 19.94 |
| remove final 20% | 0.2777 | 18.33 | 0.2792 | 20.68 |
| remove final 30% | 0.2777 | 19.18 | 0.2792 | 21.65 |
| boost (1.5×) final 10% | 0.2773 | 16.75 | 0.2789 | 18.39 |
| boost (1.5×) final 20% | 0.2772 | 16.51 | 0.2787 | 18.96 |
| boost (1.5×) final 30% | 0.2772 | **16.24** | 0.2790 | **18.23** |

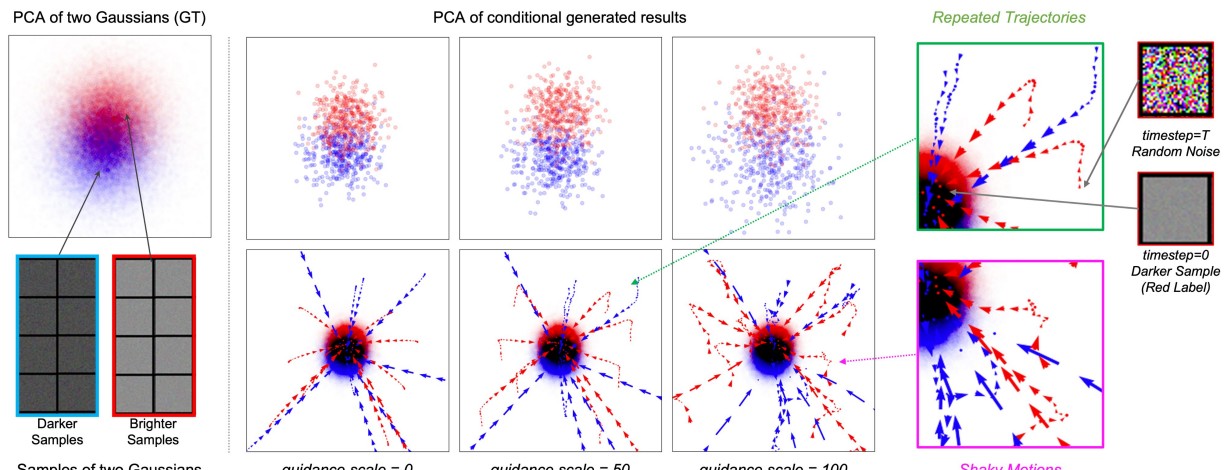

Figure 16: **Two-Gaussians Example.** We employ DDPM with CFG to fit two Gaussian distributions, a bright one (red) and a darker one (blue). The middle panel showcases samples of generation trajectories at different guidance scales $\omega$, using PCA visualization. Increasing guidance scale $\omega$ raises two issues: *repeated trajectory*: when $\omega$=50 the generation diverges from its expected direction before converging again, and *shaky motion*: when $\omega$=100 some trajectories wander aimlessly.

one sampled with low values of intensity (dark noisy images in the bottom-left of Figure 16), and the other with high-intensities (bright noisy images). The top-left part in Figure 16 shows the PCA (Kambhatla & Leen, 1997)-visualised distribution of the two sets, and the bottom-left part shows some ground-truth images. To fit these two labelled distributions, we employ DDPM (Ho et al., 2020) with *CFG* (Ho & Salimans, 2021) conditioned on intensity labels.

Upon completion of the training, we can adjust the guidance scale $\omega$ to balance between the sample fidelity and condition adherence, illustrated in the right part of Figure 16. The first row depicts the variations in generated distributions on different $\omega$ (from 0 to 100), visualized by the same PCA parameters. The second row shows the entire diffusion trajectory for sampled data points (same seeds across different $\omega$): progressing from a random sample (*i.e.*, standard Gaussian) when $t = T$ to the generated data (blue or red in Figure 16) when $t = 0$.

**Emerging issues and explainable factors.** As $\omega$ increases, the two generated distributions diverge due to *guidance term* in Eq. 7 shifting the generation towards different labels at a fidelity cost (see Figure 16 first row).

As shown in Figure 16 (second row), two issues arise: (i) *repeated trajectories* that diverge from the expected convergence path before redirecting to it; and (ii) *shaky motions* that wander along the trajectory.

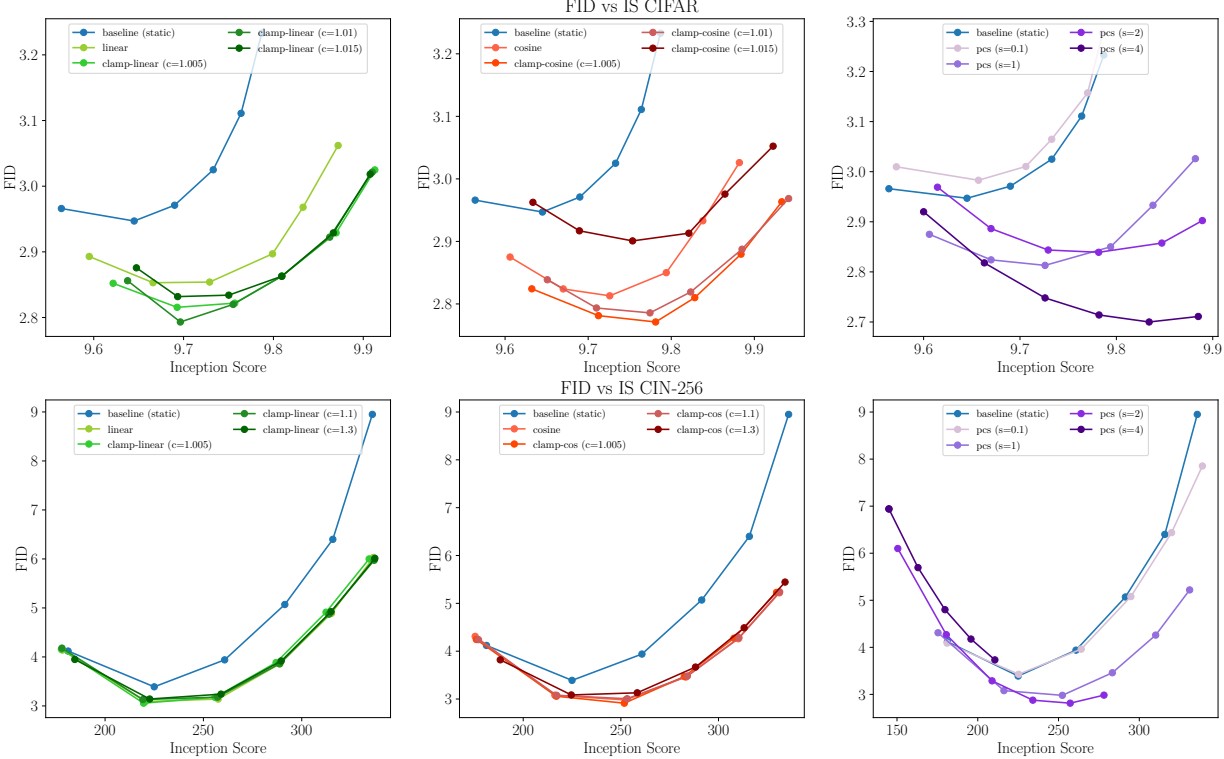

Figure 17: **Class-conditioned image generation results of two parameterized families (clamp-linear, clamp-cosine and pcs) on CIFAR-10 and CIN-256**. Optimising parameters of guidance results in performance gains, however, these parameters do not generalize across models and datasets.

These two issues can be attributed to two factors: (1) incorrect classification prediction, and (2) the conflicts between *guidance* and *generation* terms in Eq. 7. For the former, optimal guidance requires a *flawless* classifier, whether explicit for *CG* or implicit for *CFG*. In reality, discerning between two noisy data is challenging and incorrect classification may steer the generation in the wrong direction, generating shaky trajectories. A similar observation is reported in Zheng et al. (2022); Dinh et al. (2023b) for *CG* and in Li et al. (2023) for *CFG*. For the latter, due to the strong incentive of the classifier to increase the distance with respect to the other classes, trajectories often show a U-turn before gravitating to convergence (repeated trajectory in Figure 16). We argue that this anomaly is due to the conflict between *guidance* and *generation* terms in Eq. 7.

In conclusion, along the generation, the guidance can steer suboptimally (especially when $t \rightarrow T$), and even impede generation. We argue that these **erratic behaviours** contribute to the **performance dichotomy between fidelity and condition adherence** (Ho & Salimans, 2021; Dhariwal & Nichol, 2021).

## D  Comparison of Parameterized Schedulers

### D.1  Parameterized Comparison on Class-Conditioned Generation

For CIFAR-10-DDPM, we show in Figure 17 upper panels (see all data in Table 5, 6, 7) the comparison of two parameterized functions families: (i) clamp family on linear and cosine and (ii) pcs family mentioned in Gao et al. (2023).

The ImageNet-256 and Latent Diffusion Model (LDM) results are presented in Figure 17 lower panels and (data in Table 9, 10, 11).

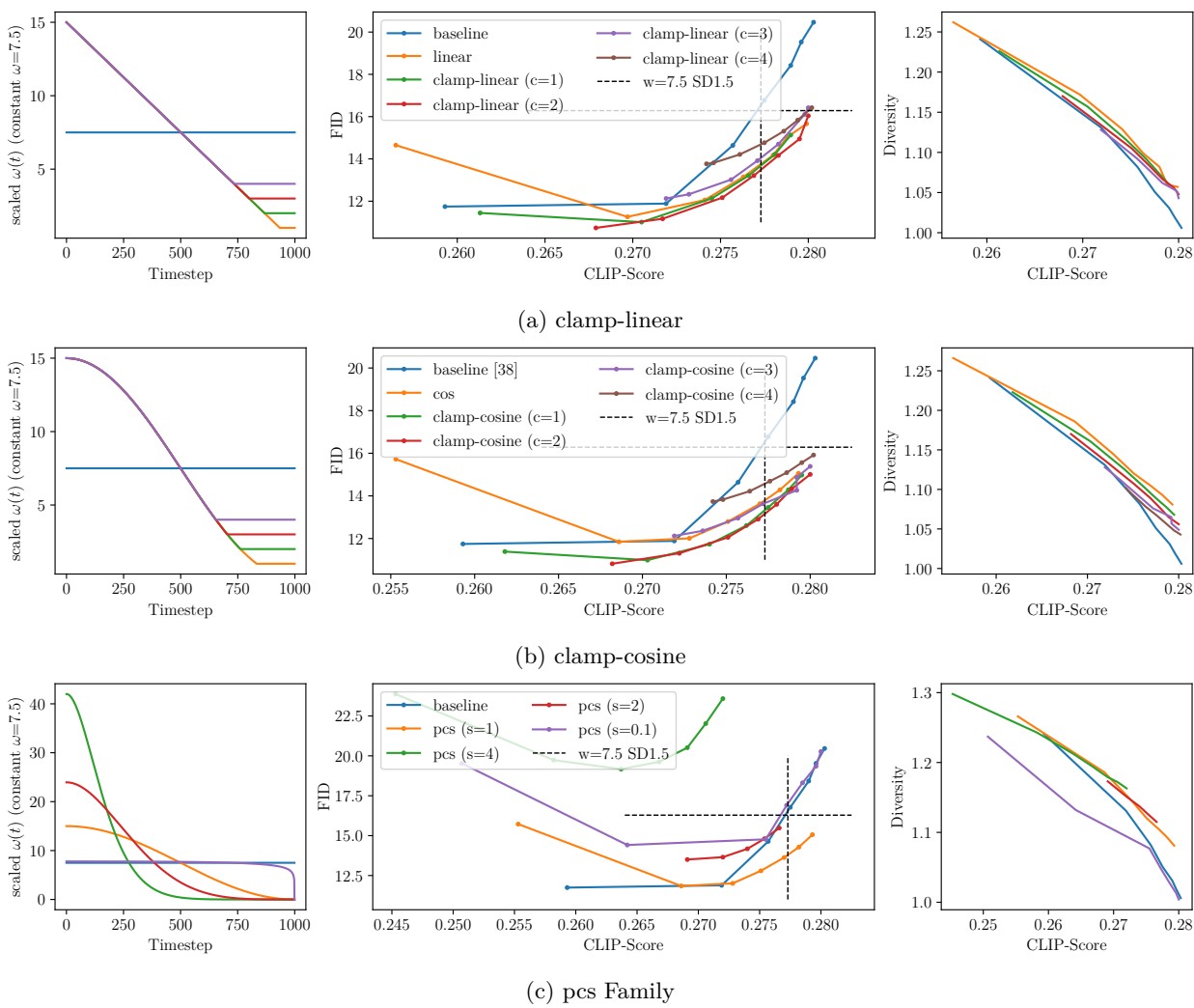

Figure 18: **Text-to-image generation FID and diversity of all two parameterized families (clamp with clamp-linear, clamp-cosine and pcs) on SD1.5** (left to right): **(a) parameterized scheduler curves**; **(b) FID vs. CS of SD1.5** and **(c) FID vs. Div. of SD1.5.** We show that in terms of diversity, the clamp family still achieves more diverse results than the baseline, though it reduces along the clamping parameter, as the beginning stage of the diffusion is muted.

The conclusion of these parts is as follows: (i) optimising both groups of parameterized function helps improve the performance of FID-CS; (ii) the optimal parameters for different models are very different and fail to generalize across models and datasets.

## D.2   Parameterized Comparison on Text-to-image Generation

We then show the FID vs. CS and Diversity vs. CS performance of the parameterized method in Figure 18. The conclusion is coherent with the main paper: all parameterized functions can enhance performance on both FID and diversity, provided that the parameters are well-selected. Moreover, for the clamp family, it appears that the clamp parameter also adjusts the degree of diversity of the generated images; lowering the clamp parameter increases the diversity. We recommend that users tune this parameter according to the specific model and task. For SDXL, the clamp-cosine is shown in Figure 19, and also reaches a similar conclusion.

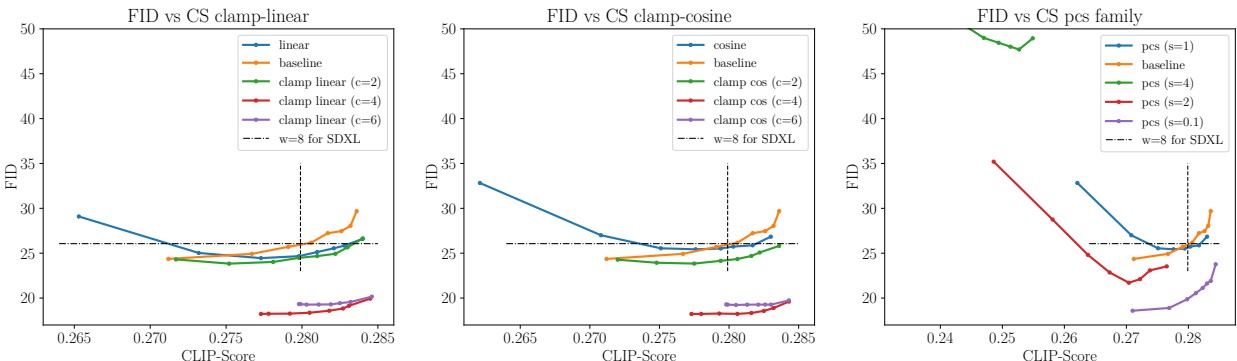

Figure 19: **Text-to-image generation results of two parameterized families (clamp-linear, clamp-cosine and pcs) on SDXL.** Both clamps reach their best FID-CS at $c = 4$ vs $s = 0.1$ for pcs, which differ from the optimal parameters for SD1.5.

# E   Qualitative Results

**More Results of Parameterized Functions on SDXL**    In Figure 20, we show more examples of different parameterized functions. It appears that carefully selecting the parameter ($c = 4$), especially for the clamp-linear method, achieves improvement in image quality in terms of composition (e.g., template), detail (e.g., cat), and realism (e.g., dog statue). However, for SDXL, this method shows only marginal improvements with the pcs family, which tends to produce images with incorrect structures and compositions, leading to fuzzy images.

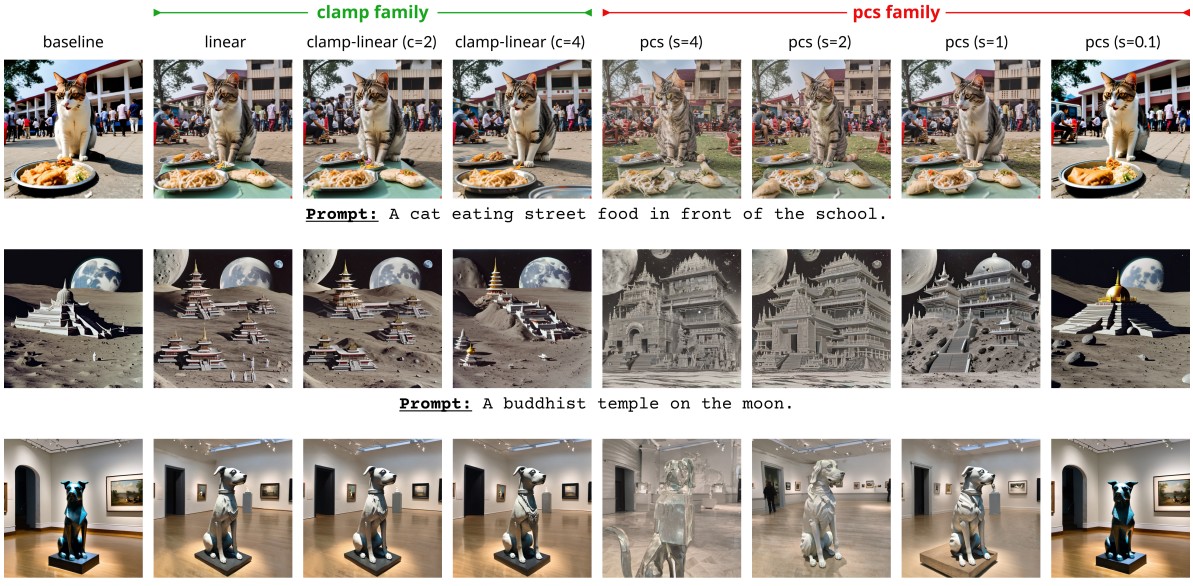

Figure 20: Qualitative comparison clamp vs. pcs family, we see clearly that clamping at $c = 4$ gives the best visual qualitative results.

**Stable Diffusion v1.5.**    Figure 21 shows qualitative results of using increasing shaped methods: linear, cosine compared against the baseline. It shows clearly that the increasingly shaped heuristic guidance generates more diversity and the baseline suffers from a collapsing problem, i.e., different sampling of the same prompt seems only to generate similar results. In some figures, e.g., Figure 21 with an example of

Table 2: **Ablation on sampling steps DDIM.** Experiment on CIN-256 and Latent Diffusion Model

| steps | baseline (static) | | linear | | cosine | |
|---|---|---|---|---|---|---|
| | FID↓ | IS↑ | FID↓ | IS↑ | FID↓ | IS↑ |
| 50 | 3.393 | 220.6 | 3.090 | 225.0 | 2.985 | 252.4 |
| 100 | **3.216** | 229.8 | **2.817** | 225.2 | 2.818 | 255.3 |
| 200 | 3.222 | 229.5 | 2.791 | 223.2 | **2.801** | 254.3 |

the mailbox, we can see that the baseline ignores graffiti and increasing heuristic guidance methods can correctly retrieve this information and illustrate it in the generated images. We also see in M&M's that heuristic guidance methods show more diversity in terms of colour and materials. with much richer variance and image composition. However some negative examples can also be found in Figure 21, in particular, the foot of horses in the prompt: a person riding a horse while the sun sets. We posit the reason for these artefacts is due to the overmuting of the initial stage and overshooting the final stage during the generation, which can be rectified by the clamping method.

**SDXL.** The SDXL (Podell et al., 2023) shows better diversity and image quality comparing to its precedent. Whereas some repetitive concepts are still present in the generated results: see Figure 22, that first row *"A single horse leaning against a wooden fence"* the baseline method generate only brown horses whereas all heuristic methods give a variety of horse colours. A similar repetitive concept can also be found in the *"A person stands on water skies in the water"* with the color of the character. For the spatial combination diversity, please refer to the example in Figure 23: *"A cobble stone courtyard surrounded by buildings and clock tower."* where we see that heuristic methods yield more view angle and spatial composition. Similar behaviour can be found in the example of *"bowl shelf"* in Figure 22 and *"teddy bear"* in Figure 22.

**SD3.** In SD3 (see Figure 24), we demonstrate that the schedulers enhance the details (e.g., books, street, lake surface), improve tone and chromatic performance (e.g., street and sunset), and lead to a better understanding of the prompt (e.g., giant clock).

## F  Ablation on Robustness and Generalization

**Different DDIM steps.** DDIM sampler allows for accelerated sampling (e.g., 50 steps as opposed to 1000) with only a marginal compromise in generation performance. In this ablation study, we evaluate the effectiveness of our dynamic weighting schedulers across different sampling steps. We use the CIN256-LDM codebase, with the same configuration as our prior experiments of class-conditioned generation. We conduct tests with $50, 100$, and $200$ steps, for baseline and two heuristics (linear and cosine), all operating at their optimal guidance scale in Tab 8. The results, FID vs. IS for each sampling step, are presented in Tab. 2. We observe that the performance of dynamic weighting schedulers remains stable across different timesteps.

**Different Solvers.** To validate the generalizability of our proposed method beyond the DDIM (Song et al., 2020) sampler used in the experiment Section, we further evaluated its performance using the more advanced DPM-Solver (Lu et al., 2022a) sampler (3rd order). This sampler is capable of facilitating diffusion generation with fewer steps and enhanced efficiency compared to DDIM. The experiment setup is similar to the text-to-image generation approach using Stable Diffusion (Rombach et al., 2022) v1.5. The results of this experiment are reported in Table 3 and visually illustrated in Figure 25.

As depicted in Figure 25: our proposed methods continue to outperform the baseline (static guidance) approach. Substantial improvements are seen in both FID and CLIP-Score metrics, compared to baseline (w=7.5) for example. Notably, these gains become more pronounced as the guidance weight increases, a trend that remains consistent with all other experiments observed across the paper.

**Diversity** Diversity plays a pivotal role in textual-based generation tasks. Given similar text-image matching levels (usually indicated by CLIP-Score), higher diversity gives users more choices of generated content. Most applications require higher diversity to prevent the undesirable phenomenon of content collapsing,

baseline          linear          cosine

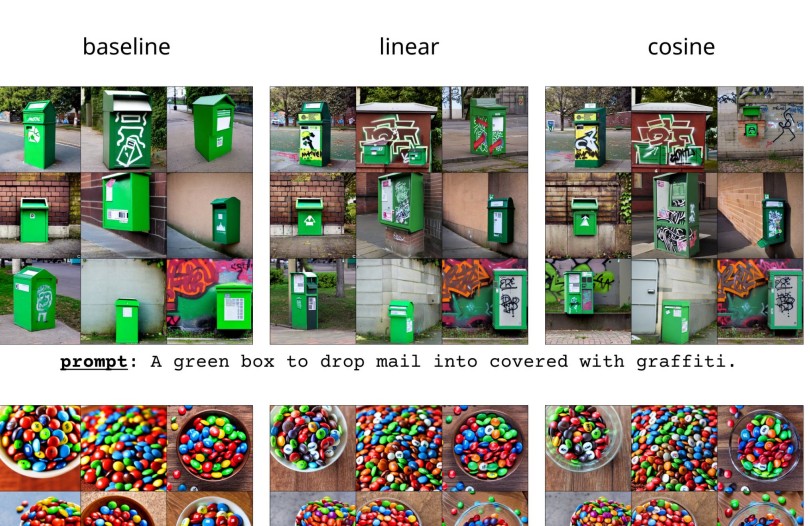

**prompt**: A green box to drop mail into covered with graffiti.

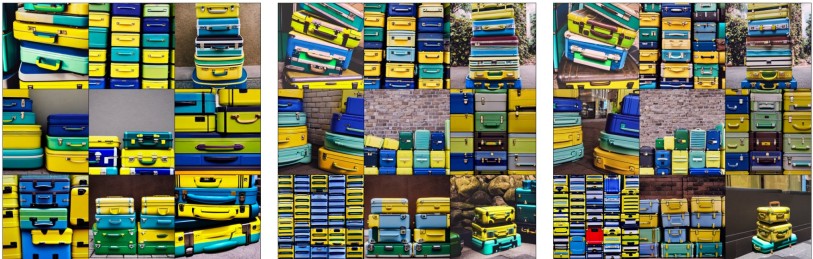

**prompt**:  Different colors of M&M's in a bowl on the table

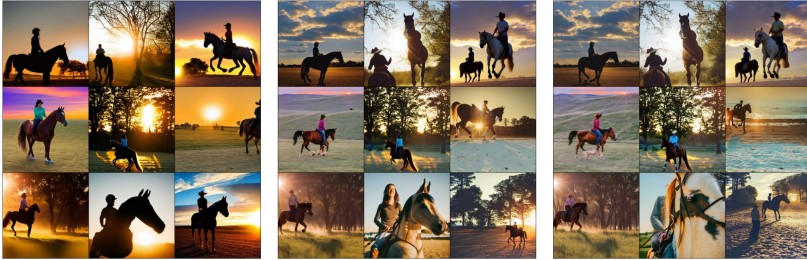

**prompt**: Yellow, blue, and green suitcases stacked on each other.

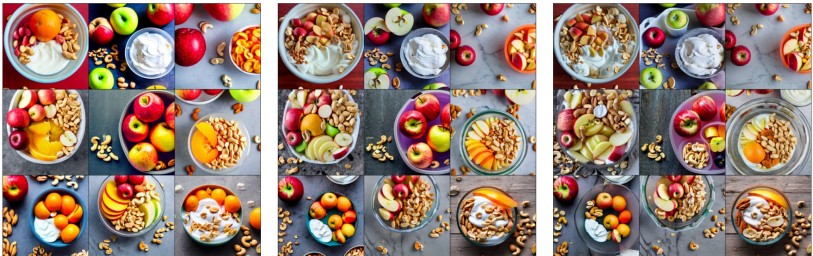

**prompt**: Person riding a horse while the sun sets.

**prompt**: An apple, orange, yogurt, and peanuts that are sitting
in a plastic bowl.

Figure 21: Qualitative SD1.5

baseline     linear     cosine

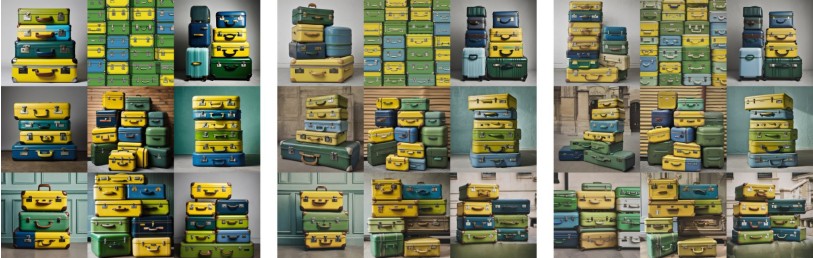

**prompt**: A single horse leaning against a wooden fence.

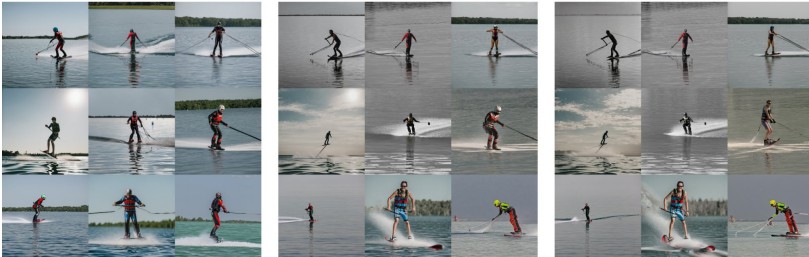

**prompt**: Yellow, blue, and green suitcases stacked on each other.

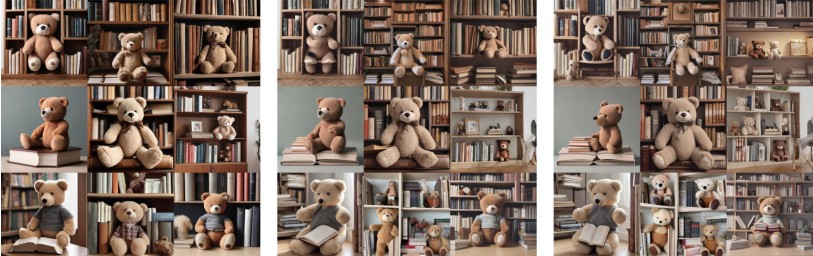

**prompt**: A person stands on water skis in the water.

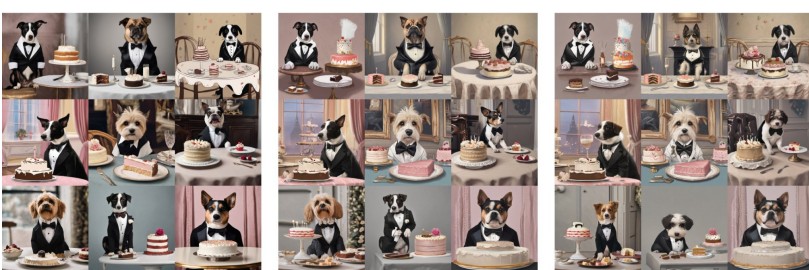

**prompt**: There is a stuffed bear sitting on a book shelf

**prompt**: A dog in a tuxedo is sitting at a table on which a piece of cake is sitting.

Figure 22: Qualitative SDXL (1)

baseline          linear          cosine

**prompt**: A cobble stone courtyard surrounded by buildings and a clock tower.

**prompt**:   A man is skateboarding across a city street.

**prompt**: A shelf with bowls lined up on it.

**prompt**: A double decker bus is moving past a tall building.

**prompt**: A bathroom with teal tiles and several sinks.

Figure 23: Qualitative SDXL (2)

**baseline**  **linear**  **linear (c=0.5)**  **linear (c=1)**

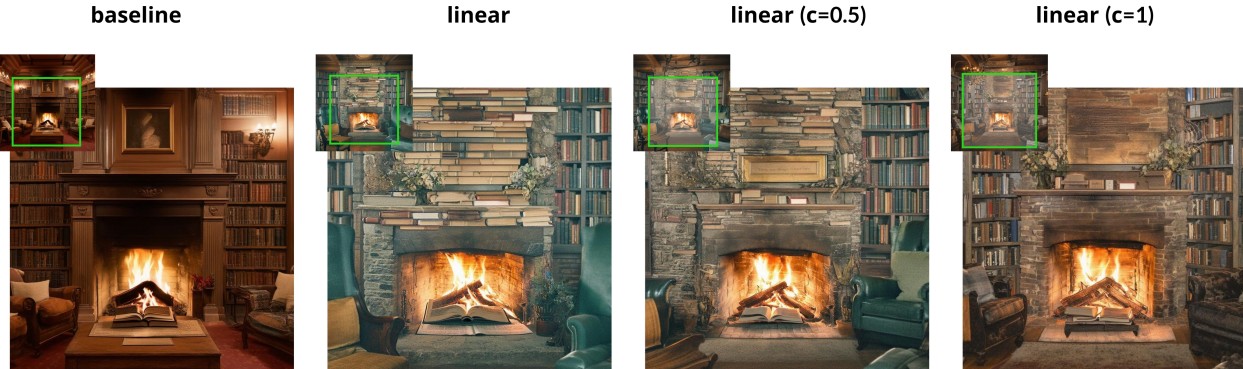

**prompt**: A cozy library filled with ancient books, warm lighting, and a crackling fireplace.

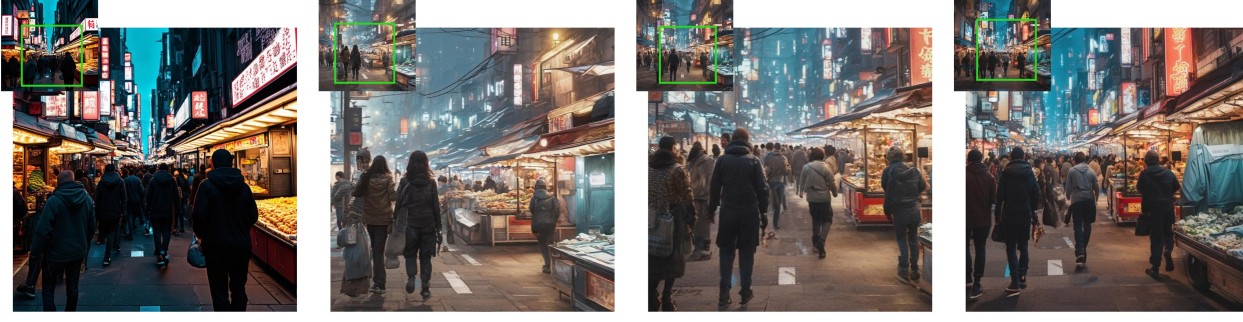

**prompt**: A cyberpunk street scene at night, filled with glowing signs, street vendors, and bustling crowds.

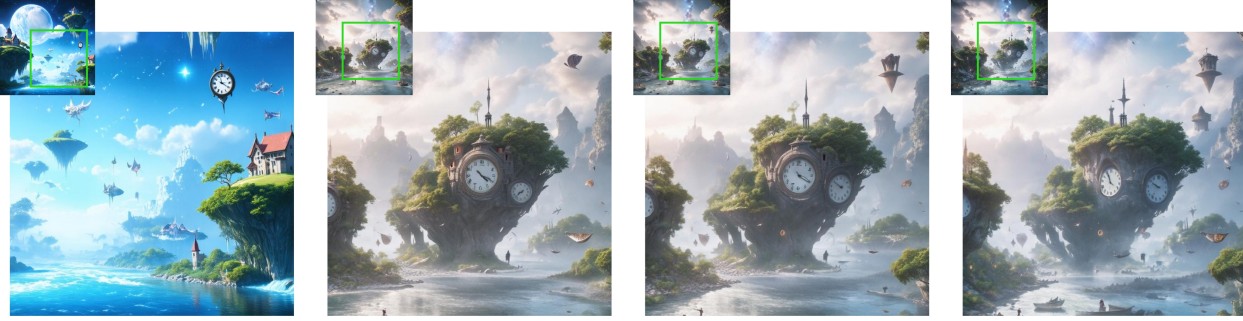

**prompt**: A surreal dreamscape with floating islands, giant clocks, and a river of stars.

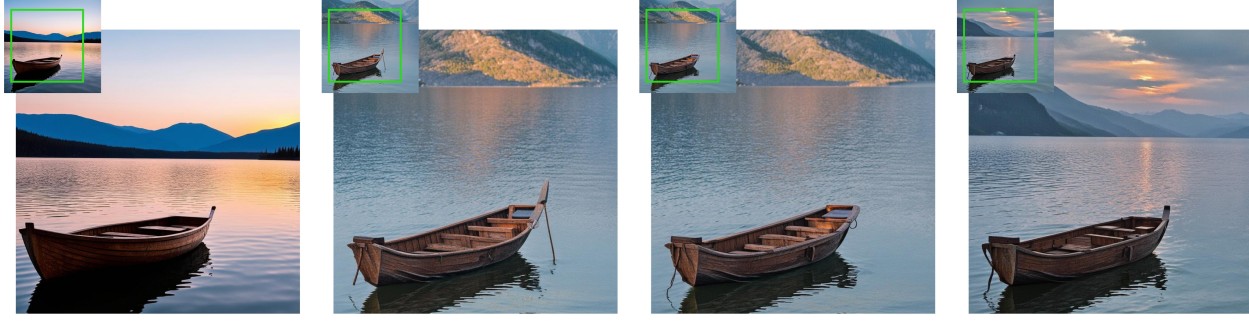

**prompt**: A serene lake at sunset with mountains in the distance and a small wooden boat floating on the water.

Figure 24: Qualitative SD3

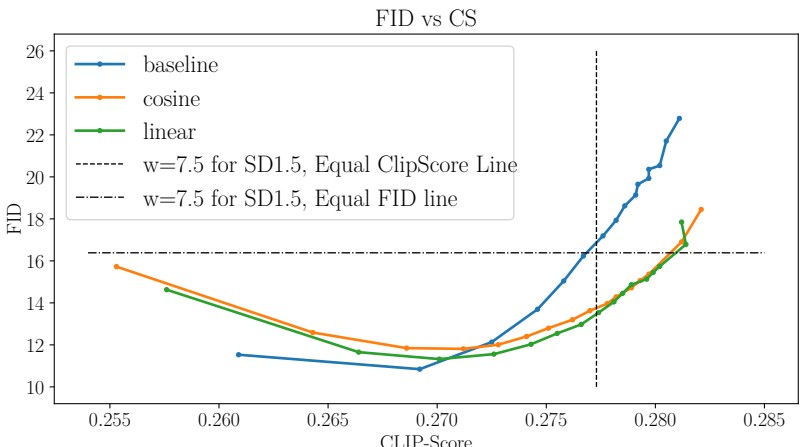

Figure 25: **FID vs. CLIP-Score** generated by SD1.5 (Rombach et al., 2022) with DPM-Solver (Lu et al., 2022a)

Table 3: Experiment of FID and CLIP-Score generated by Stable Diffusion v1.5 with DPM-Solver Lu et al. (2022a)

|  | w | 1 | 3 | 5 | 7 | 9 | 11 | 13 | 15 | 20 |
|---|---|---|---|---|---|---|---|---|---|---|
| *baseline(static)* | clip-score | 0.2287 | 0.2692 | 0.2746 | 0.2767 | 0.2782 | 0.2791 | 0.2797 | 0.2802 | 0.2805 |
|  | FID | 28.188 | 10.843 | 13.696 | 16.232 | 17.933 | 19.136 | 19.930 | 20.538 | 21.709 |
| *linear* | clip-score | 0.2287 | 0.2646 | 0.2713 | 0.2743 | 0.2762 | 0.2774 | 0.2785 | 0.2792 | 0.2813 |
|  | FID | 28.188 | 13.032 | 11.826 | 12.181 | 12.830 | 13.461 | 13.984 | 14.541 | 15.943 |
| *cosine* | clip-score | 0.2287 | 0.2643 | 0.2712 | 0.2741 | 0.2762 | 0.2778 | 0.2789 | 0.2797 | 0.2812 |
|  | FID | 28.188 | 12.587 | 11.810 | 12.400 | 13.197 | 13.968 | 14.717 | 15.366 | 16.901 |

where multiple samplings of the same prompt yield nearly identical or very similar results. We utilize the standard deviation within the image embedding space as a measure of diversity. This metric can be derived using models such as Dino-v2 Oquab et al. (2023) or CLIP Radford et al. (2021). Figure 26 provides a side-by-side comparison of diversities computed using both Dino-v2 and CLIP, numerical results are also reported in Table. 16. It is evident that Dino-v2 yields more discriminative results compared to the CLIP embedding. While both embeddings exhibit similar trends, we notice that CLIP occasionally produces a narrower gap between long captions (-L) and short captions (-S). In some instances, as depicted in Figure 26, CLIP even reverses the order, an observation not apparent with the Dino-v2 model. In both cases, our methods are consistently outperforming the baseline on both metrics.

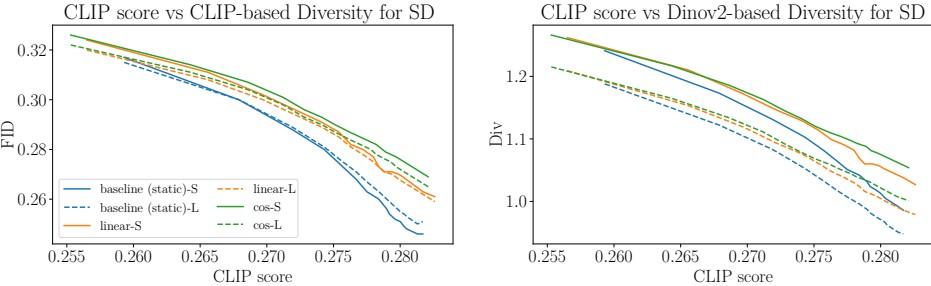

Figure 26: **Experiment on Stable Diffusion on two types of diversity.** Zero-shot COCO 10k CLIP-Score vs. Diversity computed by CLIP and Dino-v2 respectively.

## G  Detailed Table of Experiments

In this section, we show detailed tables of all experiments relevant to the paper:

- **CIFAR-10-DDPM:** results of different shapes of heuristics (Table 4), results of parameterized methods (Table 5, Table 6, Table 7)

- **CIN (ImageNet) 256-LDM:** results of different shapes of heuristics (Table 8) and results of parameterized methods (Table 9, Table 9, Table 11)

- **Stable Diffusion 1.5:** results of different shapes of heuristics in Table 12 and results of parameterized methods in Table 13.

- **Stable Diffusion XL:** results of different shapes of heuristics in Table 14 and results of parameterized methods in Table 15.

Table 4: **Experiment of different Heuristics on CIFAR-10 DDPM.** We evaluate the FID and IS results for the baseline, all heuristic methods (green for increasing, red for decreasing and purple for non-linear) of 50K images. Best FID and IS are highlighted. We see clearly that the increasing shapes outperform all the others.

| Guidance Scale | baseline (static) | | linear | | cos | | invlinear | | sin | | Λ-shape | | V-shape | |
|---|---|---|---|---|---|---|---|---|---|---|---|---|---|---|
| | FID | IS | FID | IS | FID | IS | FID | IS | FID | IS | FID | IS | FID | IS |
| 1.10 | 2.966 | 9.564 | 2.893 | 9.595 | 2.875 | 9.606 | **3.033** | 9.554 | **3.068** | 9.550 | **3.017** | 9.615 | 3.005 | 9.550 |
| 1.15 | 2.947 | 9.645 | 2.854 | 9.666 | 2.824 | 9.670 | 3.050 | 9.628 | 3.086 | 9.612 | 3.040 | 9.698 | 2.954 | 9.596 |
| 1.20 | **2.971** | 9.690 | 2.854 | 9.729 | **2.813** | 9.726 | 3.106 | 9.643 | 3.149 | 9.645 | 3.119 | 9.738 | **2.928** | 9.644 |
| 1.25 | 3.025 | 9.733 | 2.897 | 9.799 | 2.850 | 9.794 | 3.192 | 9.675 | 3.261 | 9.660 | 3.251 | 9.746 | 2.930 | 9.677 |
| 1.30 | 3.111 | 9.764 | 2.968 | 9.833 | 2.933 | 9.838 | 3.311 | 9.689 | 3.389 | 9.664 | 3.407 | 9.774 | 2.951 | 9.725 |
| 1.35 | 3.233 | **9.787** | 3.062 | **9.872** | 3.026 | **9.882** | 3.460 | **9.700** | 3.543 | **9.678** | 3.606 | **9.804** | 2.985 | **9.763** |

## H  User Study

In this section, we elaborate on the specifics of our user study setup corresponding to Figure 3. (b) in our main manuscript.

Table 5: **Experiment of clamp-linear on CIFAR-10 DDPM.** We evaluate the FID and IS results for the baseline, parameterized method as clamp-linear of 50K images FID. Best FID and IS are highlighted, the optimal parameter seems at $c = 1.1$.

| Guidance Scale | baseline (static) | | linear | | linear (c=1.05) | | linear (c=1.1) | | linear (c=1.15) | |
|---|---|---|---|---|---|---|---|---|---|---|
| | FID | IS | FID | IS | FID | IS | FID | IS | FID | IS |
| 1.10 | 2.966 | 9.564 | 2.893 | 9.595 | 2.852 | 9.622 | 2.856 | 9.638 | 2.876 | 9.647 |
| 1.15 | **2.947** | 9.645 | **2.853** | 9.666 | **2.816** | 9.693 | **2.793** | 9.696 | **2.832** | 9.693 |
| 1.20 | 2.971 | 9.690 | 2.854 | 9.729 | 2.822 | 9.757 | 2.820 | 9.755 | 2.834 | 9.750 |
| 1.25 | 3.025 | 9.733 | 2.897 | 9.799 | 2.863 | 9.809 | 2.863 | 9.809 | 2.863 | 9.809 |
| 1.30 | 3.111 | 9.764 | 2.968 | 9.833 | 2.929 | 9.870 | 2.922 | 9.863 | 2.929 | 9.867 |
| 1.35 | 3.233 | 9.787 | 3.062 | 9.872 | 3.025 | 9.913 | 3.021 | 9.910 | 3.018 | 9.908 |

Table 6: **Experiment of clamp-cosine on CIFAR-10 DDPM.** We evaluate the FID and IS results for the baseline, parameterized method as clamping on the cosine increasing heuristic (clamp-cosine) of 50K images. Best FID and IS are highlighted. It sees the optimising clamping parameter helps to improve the FID-IS performance, the optimal parameter seems at $c = 1.05$.

| Guidance Scale | baseline (static) | | cos | | cos (c=1.05) | | cos (c=1.1) | | cos (c=1.15) | |
|---|---|---|---|---|---|---|---|---|---|---|
| | FID | IS | FID | IS | FID | IS | FID | IS | FID | IS |
| 1.10 | 2.966 | 9.564 | 2.875 | 9.606 | 2.824 | 9.632 | 2.839 | 9.651 | 2.963 | 9.633 |
| 1.15 | **2.947** | 9.645 | 2.824 | 9.670 | 2.781 | 9.712 | 2.794 | 9.710 | 2.917 | 9.689 |
| 1.20 | 2.971 | 9.690 | **2.813** | 9.726 | **2.771** | 9.781 | **2.786** | 9.774 | **2.901** | 9.753 |
| 1.25 | 3.025 | 9.733 | 2.850 | 9.794 | 2.810 | 9.828 | 2.819 | 9.823 | 2.913 | 9.821 |
| 1.30 | 3.111 | 9.764 | 2.933 | 9.838 | 2.880 | 9.884 | 2.888 | 9.885 | 2.976 | 9.865 |
| 1.35 | 3.233 | 9.787 | 3.026 | 9.882 | 2.963 | 9.933 | 2.969 | 9.941 | 3.052 | 9.923 |

Table 7: **Experiment of pcs family on CIFAR-10 DDPM.** We evaluate the FID and IS results for the baseline, parameterized pcs method of 50K image FID. Best FID and IS are highlighted. It sees the optimising clamping parameter helps to improve the FID-IS performance, the optimal parameter seems at $s = 4$.

| Guidance Scale | baseline (static) | | pcs (s=4) | | pcs (s=2) | | pcs (s=1) | | pcs (s=0.1) | |
|---|---|---|---|---|---|---|---|---|---|---|
| | FID | IS | FID | IS | FID | IS | FID | IS | FID | IS |
| 1.10 | 2.966 | 9.564 | 2.920 | 9.600 | 2.969 | 9.614 | 2.875 | 9.606 | 3.010 | 9.572 |
| 1.15 | **2.947** | 9.645 | 2.818 | 9.663 | 2.886 | 9.670 | 2.824 | 9.670 | **2.983** | 9.657 |
| 1.20 | 2.971 | 9.690 | 2.748 | 9.726 | **2.844** | 9.729 | **2.813** | 9.726 | 3.010 | 9.706 |
| 1.25 | 3.025 | 9.733 | 2.714 | 9.782 | 2.839 | 9.782 | 2.850 | 9.794 | 3.065 | 9.733 |
| 1.30 | 3.111 | 9.764 | **2.700** | 9.834 | 2.858 | 9.847 | 2.933 | 9.838 | 3.157 | 9.770 |
| 1.35 | 3.233 | 9.787 | 2.711 | 9.885 | 2.902 | 9.889 | 3.026 | 9.882 | 3.276 | 9.786 |

Table 8: **Experiment of different Heuristics on CIN-256-LDM.** We evaluate the FID and IS results for the baseline, all heuristic methods (green for increasing, red for decreasing and purple for non-linear) of 50K images FID. Best FID and IS are highlighted. We see clearly that the increasing shapes outperform all the others.

| guidance | baseline | | linear | | cos | | invlinear | | sin | | Λ-shape | | V-shape | |
|---|---|---|---|---|---|---|---|---|---|---|---|---|---|---|
| | FID | IS | FID | IS | FID | IS | FID | IS | FID | IS | FID | IS | FID | IS |
| 1.4 | 4.117 | 181.2 | 4.136 | 178.3 | 4.311 | 175.4 | 4.323 | 180.7 | 4.405 | 180.2 | **3.444** | 207.8 | 6.118 | 146.2 |
| 1.6 | **3.393** | 225.0 | **3.090** | 220.6 | 3.083 | 216.2 | **3.974** | 222.7 | **4.176** | 221.7 | 3.694 | 256.5 | 4.450 | 176.8 |
| 1.8 | 3.940 | 260.8 | 3.143 | 257.5 | **2.985** | 252.4 | 4.797 | 257.3 | 5.087 | 254.8 | 4.922 | 294.9 | **3.763** | 206.1 |
| 2.0 | 5.072 | 291.4 | 3.858 | 288.9 | 3.459 | 283.3 | 6.085 | 284.2 | 6.398 | 281.2 | 6.517 | 324.8 | 3.806 | 232.2 |
| 2.2 | 6.404 | 315.8 | 4.888 | 315.1 | 4.256 | 310.1 | 7.517 | 306.9 | 7.835 | 303.4 | 8.164 | 346.2 | 4.293 | 255.7 |
| 2.4 | 8.950 | **335.9** | 6.032 | **336.5** | 5.215 | **331.2** | 8.978 | 325.5 | 9.291 | 321.3 | 9.664 | 362.9 | 5.051 | 277.0 |

Table 9: **Experiment of clamp-linear family on CIN-256-LDM.** We evaluate the FID and IS results for the baseline, parameterized clamp-linear on 50K images FID. Best FID and IS are highlighted. It sees the optimising parameter helps to improve the FID-IS performance, the optimal parameter seems at $c = 1.005$.

| guidance | baseline | | linear | | linear (c=1.005) | | linear (c=1.1) | | linear (c=1.3) | |
|---|---|---|---|---|---|---|---|---|---|---|
| | FID | IS | FID | IS | FID | IS | FID | IS | FID | IS |
| 1.4 | 4.12 | 181.2 | 4.14 | 178.3 | 4.16 | 177.8 | 4.18 | 178.1 | 3.95 | 184.6 |
| 1.6 | **3.39** | 225.0 | **3.09** | 220.6 | **3.06** | 219.6 | **3.13** | 219.2 | **3.14** | 222.7 |
| 1.8 | 3.94 | 260.8 | 3.14 | 257.5 | 3.18 | 255.9 | 3.18 | 257.2 | 3.24 | 259.0 |
| 2.0 | 5.07 | 291.4 | 3.86 | 288.9 | 3.88 | 287.0 | 3.86 | 288.7 | 3.92 | 289.6 |
| 2.2 | 6.40 | 315.8 | 4.89 | 315.1 | 4.91 | 312.4 | 4.87 | 313.8 | 4.92 | 314.9 |
| 2.4 | 8.95 | **335.9** | 6.03 | **336.5** | 6.00 | **334.3** | 5.97 | **336.8** | 6.01 | **337.2** |

Table 10: **Experiment of clamp-cosine family on CIN-256-LDM.** We evaluate the FID and IS results for the baseline, parameterized method of clamp-cosine method on 50K images. Best FID and IS are highlighted. It sees the optimising parameter helps to improve the FID-IS performance, the optimal parameter seems at $c = 1.005$.

| guidance | baseline | | cosine | | cosine (c=1.005) | | cosine (c=1.1) | | cosine (c=1.3) | |
|---|---|---|---|---|---|---|---|---|---|---|
| | FID | IS | FID | IS | FID | IS | FID | IS | FID | IS |
| 1.4 | 4.12 | 181.24 | 4.31 | 175.4 | 4.24 | 176.0 | 4.24 | 177.1 | 3.82 | 188.2 |
| 1.6 | **3.39** | 224.96 | 3.08 | 216.2 | 3.06 | 217.0 | 3.08 | 217.1 | **3.09** | 224.6 |
| 1.8 | 3.94 | 260.85 | **2.98** | 252.4 | **2.91** | 251.8 | **3.01** | 253.2 | 3.13 | 258.4 |
| 2.0 | 5.07 | 291.37 | 3.46 | 283.3 | 3.47 | 282.5 | 3.48 | 284.1 | 3.67 | 288.2 |
| 2.2 | 6.40 | 315.84 | 4.26 | 310.1 | 4.27 | 307.9 | 4.28 | 310.5 | 4.49 | 313.1 |
| 2.4 | 8.95 | **335.86** | 5.22 | **331.2** | 5.23 | **329.7** | 5.24 | **331.3** | 5.44 | **334.1** |

Table 11: **Experiment of pcs family on CIN-256-LDM.** We evaluate the FID and IS results for the baseline, parameterized method of the pcs family of 50K images. Best FID and IS are highlighted. It sees the optimising parameter helps to improve the FID-IS performance, the optimal parameter seems at $s = 2$ for FID. Interestingly, the pcs family presents a worse IS metric, than baseline and clamp-linear/cosine methods.

| guidance | baseline | | pcs (s=4) | | pcs (s=2) | | pcs (s=1) | | pcs (s=0.1) | |
|---|---|---|---|---|---|---|---|---|---|---|
| | FID | IS | FID | IS | FID | IS | FID | IS | FID | IS |
| 1.4 | 4.12 | 181.24 | 6.94 | 144.98 | 6.10 | 150.49 | 4.31 | 175.40 | 4.09 | 181.00 |
| 1.6 | **3.39** | 224.96 | 5.69 | 162.99 | 4.27 | 180.52 | 3.08 | 216.21 | **3.43** | 225.31 |
| 1.8 | 3.94 | 260.85 | 4.80 | 179.71 | 3.29 | 208.86 | **2.98** | 252.37 | 3.96 | 264.03 |
| 2.0 | 5.07 | 291.37 | 4.18 | 195.75 | 2.88 | 234.09 | 3.46 | 283.32 | 5.08 | 294.77 |
| 2.2 | 6.40 | 315.84 | **3.73** | 210.60 | **2.81** | 257.22 | 4.26 | 310.14 | 6.44 | 319.97 |
| 2.4 | 8.95 | **335.86** | 3.457 | **224.4** | 2.98 | **278.14** | 5.22 | **331.17** | 7.85 | **339.05** |

Table 12: **Different Heuristic Modes of SD1.5**, we show FID vs. CLIP-score of 10K images. we highlight different range of clip-score by low ($\sim 0.272$), mid ($\sim 0.277$) and high ($\sim 0.280$) by pink, orange and blue colors. We see that increasing modes demonstrate the best performance at high w, whereas decreasing modes regress on the performance. non-linear modes, especially $\Lambda$-shape also demonstrate improved performance to baseline but worse than increasing shapes.

| | w | 2 | 4 | 6 | 8 | 10 | 12 | 14 |
|---|---|---|---|---|---|---|---|---|
| baseline | clip-score | 0.2593 | 0.2719 | 0.2757 | 0.2775 | 0.2790 | 0.2796 | 0.2803 |
| | FID | 11.745 | 11.887 | 14.639 | 16.777 | 18.419 | 19.528 | 20.462 |
| linear | clip-score | 0.2565 | 0.2697 | 0.2741 | 0.2763 | 0.2780 | 0.2788 | 0.2799 |
| | FID | 14.649 | 11.260 | 12.056 | 13.147 | 14.179 | 15.032 | 15.663 |
| cos | clip-score | 0.2553 | 0.2686 | 0.2728 | 0.2751 | 0.2770 | 0.2782 | 0.2793 |
| | FID | 15.725 | 11.846 | 12.009 | 12.796 | 13.629 | 14.282 | 15.058 |
| sin | clip-score | 0.261 | 0.272 | 0.2754 | 0.2773 | 0.2780 | 0.2787 | 0.2793 |
| | FID | 10.619 | 14.618 | 18.323 | 20.829 | 22.380 | 23.534 | 24.561 |
| invlinear | clip-score | 0.2608 | 0.2723 | 0.2757 | 0.2773 | 0.2781 | 0.2789 | 0.2793 |
| | FID | 10.649 | 14.192 | 17.810 | 20.206 | 21.877 | 22.962 | 24.128 |
| $\Lambda$-shape | clip-score | 0.2603 | 0.2719 | 0.2756 | 0.2774 | 0.2785 | 0.2794 | 0.2802 |
| | FID | 11.940 | 12.106 | 14.183 | 16.100 | 17.530 | 18.663 | 19.723 |
| V-shap | clip-score | 0.2569 | 0.2706 | 0.2747 | 0.2764 | 0.2773 | 0.2783 | 0.2789 |
| | FID | 11.790 | 12.407 | 15.912 | 18.220 | 19.796 | 20.992 | 21.905 |

Table 13: **Different parameterized functions of SD1.5**, we show FID vs. CLIP-score of 10K images. we highlight different range of clip-score by low ($\sim 0.272$), mid ($\sim 0.277$) and high ($\sim 0.280$) by pink, orange and blue colors. We see that for the pcs family the optimal parameter is at $s = 1$, whereas for clamp-linear and clamp-cosine methods, they are at $c = 2$.

| | w | 2 | 4 | 6 | 8 | 10 | 12 | 14 |
|---|---|---|---|---|---|---|---|---|
| baseline | clip-score | 0.2593 | 0.2719 | 0.2757 | 0.2775 | 0.2790 | 0.2796 | 0.2803 |
| | FID | 11.745 | 11.887 | 14.639 | 16.777 | 18.419 | 19.528 | 20.462 |
| pcs (s=4) | clip-score | 0.2453 | 0.2582 | 0.2637 | 0.2668 | 0.2691 | 0.2706 | 0.2720 |
| | FID | 23.875 | 19.734 | 19.167 | 19.627 | 20.513 | 22.022 | 23.585 |
| pcs (s=2) | clip-score | 0.2591 | 0.2642 | 0.2691 | 0.2720 | 0.2740 | 0.2754 | 0.2766 |
| | FID | 18.026 | 14.414 | 13.503 | 13.652 | 14.175 | 14.806 | 15.480 |
| pcs (s=1) | clip-score | 0.2553 | 0.2686 | 0.2728 | 0.2751 | 0.2770 | 0.2782 | 0.2793 |
| | FID | 15.725 | 11.846 | 12.009 | 12.796 | 13.629 | 14.282 | 15.058 |
| pcs (s=0.1) | clip-score | 0.2507 | 0.2642 | 0.2755 | 0.2772 | 0.2785 | 0.2796 | 0.2800 |
| | FID | 19.532 | 14.414 | 14.770 | 16.901 | 18.312 | 19.349 | 20.271 |
| linear (c=1) | clip-score | 0.2613 | 0.2705 | 0.2745 | 0.2766 | 0.2781 | 0.2790 | 0.2798 |
| | FID | 11.4448 | 11.011 | 12.130 | 13.211 | 14.219 | 15.129 | 15.888 |
| linear (c=2) | clip-score | 0.2679 | 0.2717 | 0.2751 | 0.2769 | 0.2783 | 0.2795 | 0.2800 |
| | FID | 10.7382 | 11.169 | 12.168 | 13.211 | 14.166 | 14.946 | 16.041 |
| linear (c=3) | clip-score | 0.2719 | 0.2732 | 0.2756 | 0.2771 | 0.2783 | 0.2798 | 0.2800 |
| | FID | 12.1284 | 12.328 | 13.019 | 13.916 | 14.701 | 16.109 | 16.420 |
| linear (c=4) | clip-score | 0.2742 | 0.2746 | 0.2761 | 0.2775 | 0.2786 | 0.2794 | 0.2802 |
| | FID | 13.768 | 13.813 | 14.213 | 14.765 | 15.311 | 15.834 | 16.422 |
| cos (c=1) | clip-score | 0.2618 | 0.2703 | 0.2740 | 0.2762 | 0.2775 | 0.2787 | 0.2795 |
| | FID | 11.386 | 10.986 | 11.732 | 12.608 | 13.460 | 14.288 | 14.978 |
| cos (c=2) | clip-score | 0.2682 | 0.2722 | 0.2751 | 0.2769 | 0.2780 | 0.2789 | 0.2800 |
| | FID | 10.816 | 11.309 | 12.055 | 12.908 | 13.602 | 14.326 | 15.008 |
| cos (c=3) | clip-score | 0.2719 | 0.2736 | 0.2757 | 0.2772 | 0.2792 | 0.2792 | 0.2800 |
| | FID | 12.121 | 12.363 | 12.956 | 13.631 | 14.263 | 14.869 | 15.385 |
| cos (c=4) | clip-score | 0.2742 | 0.2748 | 0.2764 | 0.2776 | 0.2786 | 0.2795 | 0.2802 |
| | FID | 13.734 | 13.827 | 14.222 | 14.690 | 15.090 | 15.560 | 15.916 |

Table 14: **Different Heuristic Modes of SDXL**, we show FID vs. CLIP-score of 10K images. we highlight different range of clip-score by low ($\sim 0.2770$), mid ($\sim 0.280$) and high ($\sim 0.2830$) by pink, orange and blue colors. We see that increasing modes demonstrate the best performance at high w, whereas decreasing modes regress on the performance. non-linear modes, especially $\Lambda$-shape demonstrate improved performance against baseline but regress fast when the $\omega$ is high.

| | w | 1 | 3 | 5 | 7 | 9 | 11 | 13 | 15 | 20 |
|---|---|---|---|---|---|---|---|---|---|---|
| baseline | clip-score | 0.2248 | 0.2712 | 0.2767 | 0.2791 | 0.2806 | 0.2817 | 0.2826 | 0.2832 | 0.2836 |
| | FID | 59.2480 | 24.3634 | 24.9296 | 25.7080 | 26.1654 | 27.2308 | 27.4628 | 28.0538 | 29.6868 |
| linear | clip-score | 0.2248 | 0.2653 | 0.2732 | 0.2773 | 0.2798 | 0.2810 | 0.2821 | 0.2828 | 0.2840 |
| | FID | 59.2480 | 29.0917 | 25.0276 | 24.4500 | 24.6705 | 25.1286 | 25.5488 | 25.8457 | 26.5993 |
| cosine | clip-score | 0.2248 | 0.2621 | 0.2708 | 0.2751 | 0.2776 | 0.2794 | 0.2803 | 0.2817 | 0.2830 |
| | FID | 59.2480 | 32.8264 | 27.0004 | 25.5468 | 25.4331 | 25.5244 | 25.7375 | 25.8758 | 26.8427 |
| invlinear | clip-score | 0.2248 | 0.2739 | 0.2783 | 0.2800 | 0.2814 | 0.2826 | 0.2823 | 0.2807 | 0.2730 |
| | FID | 59.2480 | 23.8196 | 25.4335 | 26.1458 | 27.8969 | 29.6194 | 31.8970 | 35.2600 | 47.8467 |
| sin | clip-score | 0.2248 | 0.2741 | 0.2786 | 0.2803 | 0.2816 | 0.2823 | 0.2816 | 0.2794 | 0.2713 |
| | FID | 59.2480 | 23.9147 | 25.4203 | 26.3137 | 28.1756 | 29.3571 | 30.5314 | 36.3049 | 51.6672 |
| $\Lambda$-shape | clip-score | 0.2248 | 0.2721 | 0.2782 | 0.2809 | 0.2826 | 0.2831 | 0.2837 | 0.2846 | 0.2849 |
| | FID | 59.2480 | 22.3927 | 24.0785 | 25.6845 | 26.7019 | 27.5095 | 28.2058 | 32.1870 | 34.9896 |
| V-shape | clip-score | 0.2248 | 0.2688 | 0.2747 | 0.2770 | 0.2785 | 0.2793 | 0.2795 | 0.2786 | 0.2736 |
| | FID | 59.2480 | 21.6560 | 22.7042 | 23.6659 | 24.0550 | 25.4073 | 26.2993 | 27.6580 | 35.2935 |

Table 15: **Different parameterized results in SDXL**, we show FID vs. CLIP-Score of pcs family and clamp family of 10K images: pcs family records best performance at $s = 0.1$, clamp-linear and clamp-cosine strategies all record best performance at $c = 4$.

| | w | 1 | 3 | 5 | 7 | 9 | 11 | 13 | 15 | 20 |
|---|---|---|---|---|---|---|---|---|---|---|
| baseline | clip-score | 0.2248 | 0.2712 | 0.2767 | 0.2791 | 0.2806 | 0.2817 | 0.2826 | 0.2832 | 0.2836 |
| | FID | 59.2480 | 24.3634 | 24.9296 | 25.7080 | 26.1654 | 27.2308 | 27.4628 | 28.0538 | 29.6868 |
| pcs ($s = 4$) | clip-score | 0.2248 | 0.2336 | 0.2396 | 0.2440 | 0.2470 | 0.2494 | 0.2513 | 0.2527 | 0.2549 |
| | FID | 59.2480 | 55.2402 | 52.0731 | 50.3335 | 48.9980 | 48.4516 | 48.0146 | 47.7025 | 48.9481 |
| pcs ($s = 2$) | clip-score | 0.2248 | 0.2486 | 0.2581 | 0.2638 | 0.2673 | 0.2704 | 0.2722 | 0.2738 | 0.2765 |
| | FID | 59.2480 | 35.2002 | 28.7500 | 24.8120 | 22.8518 | 21.7098 | 22.1061 | 23.0833 | 23.5282 |
| pcs ($s = 1$) | clip-score | 0.2248 | 0.2621 | 0.2708 | 0.2751 | 0.2776 | 0.2794 | 0.2803 | 0.2817 | 0.2830 |
| | FID | 59.2480 | 32.8264 | 27.0004 | 25.5468 | 25.4331 | 25.5244 | 25.7375 | 25.8758 | 26.8427 |
| pcs ($s = 0.1$) | clip-score | 0.2248 | 0.2710 | 0.2769 | 0.2798 | 0.2812 | 0.2823 | 0.2830 | 0.2836 | 0.2844 |
| | FID | 59.2480 | 18.5894 | 18.8975 | 19.8658 | 20.5433 | 21.1257 | 21.6248 | 21.9118 | 23.7671 |
| linear ($c = 2$) | clip-score | 0.2248 | 0.2717 | 0.2752 | 0.2781 | 0.2798 | 0.2810 | 0.2822 | 0.2830 | 0.2840 |
| | FID | 59.2480 | 24.3084 | 23.8361 | 24.0241 | 24.4806 | 24.6759 | 24.9336 | 25.6498 | 26.6398 |
| linear ($c = 4$) | clip-score | 0.2248 | 0.2773 | 0.2778 | 0.2792 | 0.2805 | 0.2818 | 0.2827 | 0.2831 | 0.2845 |
| | FID | 59.2480 | 18.2321 | 18.2517 | 18.2678 | 18.3675 | 18.5902 | 18.8356 | 19.1395 | 19.9400 |
| linear ($c = 6$) | clip-score | 0.2248 | 0.2798 | 0.2799 | 0.2803 | 0.2811 | 0.2819 | 0.2825 | 0.2832 | 0.2846 |
| | FID | 59.2480 | 19.3309 | 19.3295 | 19.2716 | 19.2801 | 19.2955 | 19.4298 | 19.5635 | 20.1577 |
| cosine ($c = 2$) | clip-score | 0.2248 | 0.2720 | 0.2748 | 0.2775 | 0.2794 | 0.2806 | 0.2816 | 0.2822 | 0.2836 |
| | FID | 59.2480 | 24.2768 | 23.9367 | 23.8442 | 24.1493 | 24.3516 | 24.6917 | 25.0779 | 25.8126 |
| cosine ($c = 4$) | clip-score | 0.2248 | 0.2773 | 0.2780 | 0.2793 | 0.2806 | 0.2816 | 0.2825 | 0.2832 | 0.2843 |
| | FID | 59.2480 | 18.2321 | 18.2336 | 18.2764 | 18.2364 | 18.3372 | 18.5678 | 18.8925 | 19.6065 |
| cosine ($c = 6$) | clip-score | 0.2248 | 0.2798 | 0.2799 | 0.2805 | 0.2813 | 0.2821 | 0.2826 | 0.2830 | 0.2843 |
| | FID | 59.2480 | 19.2943 | 19.2701 | 19.2261 | 19.2656 | 19.2711 | 19.2743 | 19.2670 | 19.7355 |

Table 16: **Experiment on SD1.5 with Diversity measures** of 10K images, comparison between the baseline and two increasing heuristic shapes, linear and cosine.

| | w | 2 | 4 | 6 | 8 | 10 | 12 | 14 | 20 | 25 |
|---|---|---|---|---|---|---|---|---|---|---|
| baseline | clip-score | 0.2593 | 0.2719 | 0.2757 | 0.2775 | 0.2790 | 0.2796 | 0.2803 | 0.2813 | 0.2817 |
| | FID | 11.745 | 11.887 | 14.639 | 16.777 | 18.419 | 19.528 | 20.462 | 22.463 | 23.810 |
| | Div-CLIP-L | 0.315 | 0.289 | 0.275 | 0.267 | 0.260 | 0.257 | 0.254 | 0.250 | 0.251 |
| | Div-Dinov2-L | 1.188 | 1.083 | 1.033 | 1.007 | 0.987 | 0.976 | 0.967 | 0.951 | 0.948 |
| | Div-CLIP-S | 0.317 | 0.288 | 0.273 | 0.263 | 0.256 | 0.252 | 0.249 | 0.246 | 0.246 |
| | Div-Dinov2-S | 1.241 | 1.131 | 1.082 | 1.051 | 1.031 | 1.019 | 1.006 | 0.992 | 0.986 |
| linear | clip-score | 0.2565 | 0.2697 | 0.2741 | 0.2763 | 0.2780 | 0.2788 | 0.2799 | 0.2817 | 0.2826 |
| | FID | 14.649 | 11.260 | 12.056 | 13.147 | 14.179 | 15.032 | 15.663 | 17.478 | 18.718 |
| | Div-CLIP-L | 0.320 | 0.300 | 0.289 | 0.281 | 0.275 | 0.271 | 0.268 | 0.262 | 0.259 |
| | Div-Dinov2-L | 1.209 | 1.119 | 1.076 | 1.048 | 1.030 | 1.016 | 1.006 | 0.986 | 0.979 |
| | Div-CLIP-S | 0.324 | 0.302 | 0.291 | 0.282 | 0.277 | 0.271 | 0.270 | 0.263 | 0.261 |
| | Div-Dinov2-S | 1.262 | 1.172 | 1.129 | 1.099 | 1.082 | 1.060 | 1.057 | 1.038 | 1.027 |
| cos | clip-score | 0.2553 | 0.2686 | 0.2728 | 0.2751 | 0.2770 | 0.2782 | 0.2793 | 0.2812 | 0.2821 |
| | FID | 15.725 | 11.846 | 12.009 | 12.796 | 13.629 | 14.282 | 15.058 | 16.901 | 18.448 |
| | Div-CLIP-L | 0.322 | 0.304 | 0.293 | 0.287 | 0.282 | 0.278 | 0.275 | 0.268 | 0.265 |
| | Div-Dinov2-L | 1.215 | 1.134 | 1.092 | 1.068 | 1.051 | 1.039 | 1.030 | 1.008 | 1.001 |
| | Div-CLIP-S | 0.326 | 0.307 | 0.296 | 0.290 | 0.285 | 0.282 | 0.278 | 0.272 | 0.269 |
| | Div-Dinov2-S | 1.266 | 1.186 | 1.145 | 1.120 | 1.104 | 1.093 | 1.081 | 1.063 | 1.054 |

Table 17: **Experiment on SDXL with Diversity.**, we present FID vs. CLIP-Score (CS) for SDXL of 10K images, and we see the similar trending to Table 16 that the heuristic methods outperform the baseline, both on FID and Diversity.

| | w | 3 | 5 | 7 | 8 | 9 | 11 | 13 | 15 | 20 |
|---|---|---|---|---|---|---|---|---|---|---|
| baseline | clip-score | 0.2712 | 0.2767 | 0.2791 | 0.2799 | 0.2806 | 0.2817 | 0.2826 | 0.2832 | 0.2836 |
| | FID | 24.36 | 24.93 | 25.71 | 26.06 | 26.17 | 27.23 | 27.46 | 28.05 | 29.69 |
| | Div-Dinov2-L | 0.951 | 0.886 | 0.857 | 0.850 | 0.841 | 0.831 | 0.827 | 0.829 | 0.853 |
| | Div-Dinov2-S | 1.052 | 0.985 | 0.950 | 0.940 | 0.934 | 0.920 | 0.916 | 0.912 | 0.927 |
| linear | clip-score | 0.2653 | 0.2732 | 0.2773 | 0.2789 | 0.2798 | 0.2810 | 0.2821 | 0.2828 | 0.2840 |
| | FID | 29.09 | 25.03 | 24.45 | 24.52 | 24.67 | 25.13 | 25.55 | 25.85 | 26.60 |
| | Div-Dinov2-L | 0.999 | 0.949 | 0.916 | 0.904 | 0.897 | 0.881 | 0.873 | 0.863 | 0.854 |
| | Div-Dinov2-S | 1.123 | 1.064 | 1.030 | 1.018 | 1.007 | 0.989 | 0.980 | 0.973 | 0.956 |
| cosine | clip-score | 0.2621 | 0.2708 | 0.2751 | 0.2764 | 0.2776 | 0.2794 | 0.2803 | 0.2817 | 0.2830 |
| | FID | 32.83 | 27.00 | 25.55 | 25.41 | 25.43 | 25.52 | 25.74 | 25.88 | 26.84 |
| | Div-Dinov2-L | 1.017 | 0.969 | 0.941 | 0.932 | 0.922 | 0.908 | 0.899 | 0.893 | 0.879 |
| | Div-Dinov2-S | 1.143 | 1.095 | 1.066 | 1.056 | 1.045 | 1.031 | 1.020 | 1.008 | 0.994 |

For the evaluation, each participant was presented with a total of 10 image sets. Each set comprised 9 images. Within each set, three pairwise comparisons were made: linear vs. baseline, and cosine vs. baseline. Throughout the study, two distinct image sets (20 images for each method) were utilized. We carried out two tests for results generated with stable diffusion v1.5 and each image are generated to make sure that their CLIP-Score are similar.

Subsequently, participants were prompted with three questions for each comparison:

1. *Which set of images is more realistic or visually appealing?*

2. *Which set of images is more diverse?*

3. *Which set of images aligns better with the provided text description?*

In total, we recorded 54 participants with each participant responding to 90 questions. We analyzed the results by examining responses to each question individually, summarizing the collective feedback.

