# OpenReview forum: "Analysis of Classifier-Free Guidance Weight Schedulers"
_TMLR — Accepted by TMLR_

### Review · Reviewer_d2J5 · 2024-08-20

**Summary Of Contributions:**

This paper performs an empirical analysis of classifier-free guidance weight schedulers, focusing on the text-to-image setting. It considers different types of heuristic schedulers and both qualitative and quantitative evaluation. The main conclusion is that in general monotonically increasing schedulers improve performance, but more complex parameterized schedulers perform well only in specific settings and do not generalize.

**Audience:**

Yes

**Broader Impact Concerns:**

No broader impact concerns are stated, but I think that it's fine because the paper only investigates the inference behavior of existing models.

**Claims And Evidence:**

Yes

**Requested Changes:**

Critical:
- Include experiments on diffusion models with alternative conditioning, such as image-to-image (https://huggingface.co/docs/diffusers/main/en/using-diffusers/img2img)

**Strengths And Weaknesses:**

### Strengths ###

- The paper is generally well-written, with lots of illustrations.
- Diffusion models are very popular right now, both in terms of research and practical usage.
- The experiments are sound and fairly thorough.
- The paper ends by giving practical recommendation, which is useful.

### Weaknesses ###

- The paper is limited to text-to-image diffusion models, which is only a portion of the diffusion models out there. For example, adding some experiments for the case where the conditioning is different would greatly strengthen the work.
- I am not sure of the significance of the work for the research community. From the results, I think the practical value is clear, but as the authors state, previous papers have mentioned the value of increasing guidance weight schedulers.

---

> ### Author Response · Authors · 2024-10-23
>
> We appreciate the reviewer for the time and effort in reviewing our paper. In response to the reviewer’s concerns, our replies are as follows:
>
> __For weakness No.2: significance of the work for the research community:__
>
> We offer a thorough analysis of CFG schedulers: (1) While CFG schedulers have been utilized in some applications, our in-depth analysis, being __the first__ of its kind in the literature, should be seen as a strength rather than a limitation. In particular, we quantitatively demonstrate their significant impact on performance across multiple models, though it comes with a trade-off in terms of generalization, especially for the parameterized ones. (2) We also provide extensive findings and practical guidelines, aimed at assisting users in designing more effective schedulers while cautioning against potential overfitting issues.
>
> __For the change request No.1 Image-to-Image Translation (See Section 6.2 and Appendix Section A):__
>
> In our analysis, we experimented not only with text-prompt generation but also included __class-conditioned generation results in Sections 5.1 and 6.1__, where we evaluated the performance of DDPM and LDM on CIFAR10 and ImageNet separately. The conclusions align with the text-to-image generation tasks: a monotonically increasing dynamic guidance scheduler enhances generation quality, and while parameterized ones can further improve performance, it risks overfitting the parameters.
>
> As requested by the reviewer, we also included the image-to-image translation task of SD1.5 in __Section 6.2 and Appendix Section A__, in which we see clearly that linear guidance is able to improve the performance and the parameterized function has a very low influence on the generation qualities, i.e. the optimal clamping parameter is zero, which is different from the SD1.5 generation task, where the optimal parameter is found at 2. This observation also aligns with our main argument that the generalisation of the parameterized function does not exist, even for one model and different tasks.
>
> These modifications have been highlighted in __blue__ in the revised manuscript.

---

### Review · Reviewer_wiZb · 2024-10-06

**Summary Of Contributions:**

The paper conducts an in-depth analysis of guidance weights used in Classifier-Free Guidance. The authors systematically explore the role of dynamic weight schedulers, presenting empirical evidence that monotonically increasing schedulers (such as linear and cosine) outperform static guidance in fidelity and diversity. The paper also explores more complex, parameterized guidance weight schedulers, which allow for further performance improvements. These contributions provide valuable insights and practical recommendations for improving text-to-image generation performance.

**Audience:**

Yes

**Broader Impact Concerns:**

I have no broader impact concerns.

**Claims And Evidence:**

Yes

**Requested Changes:**

1. Figure 5 caption seems wrong (a and b should be swapped).
2. A theoretical analysis of the findings would be helpful.
3. I would like to see if the conclusion of the paper holds for rectified flow models (e.g., SD3).

**Strengths And Weaknesses:**

Strengths:
- The empirical analysis is thorough and comprehensive.
- The paper is clearly written and the figures are generally clear.
- The proposed solution is straightforward and effective.

Weaknesses:
- While the empirical evidences are strong, I do find a general lack of insights into why dynamic scheduler (or monotonically increasing scheduler) works better.
- It is unclear if the same conclusion holds for models not following DDPM schedule (e.g., rectified flow), which are becoming increasingly popular recently.

---

> ### Author Response · Authors · 2024-10-23
>
> We appreciate the reviewer for the time and effort in reviewing our paper. In response to the reviewer’s concerns, our replies are as follows:
>
> __For the change request No.1 Wrong caption:__
>
> We corrected the problem in the caption, and we thank the reviewer for pointing it out.
>
> __For the change request No.2 and weakness 1: Theoretical Analysis of Training Behavior and Guidance Scheduler Performance (see Section 4)__
>
> We have added a theoretical justification for classifier-free guidance during the inference process, along with a relevant discussion on the potential reasons for its strong performance (see __Section 4, “Conflicted Terms”__).
>
> Specifically, in conjunction with the toy example presented in __Appendix Section C “a toy example”__ in the revised version, we hypothesize that the effectiveness of monotonically increasing guidance (similar to certain parameterized functions) may stem from conflicts within the guidance term (i.e., the gradient of the classifier) during the early stages of the diffusion process. At this stage, the image often lacks sufficient information due to high noise levels, and an excessive guidance scale may cause “shaky” or “curvy” generation trajectories, as illustrated in __Appendix Section C__.
>
> This hypothesis is further supported by our measurements of the conflict level between the generation term and the guidance term. Additionally, we demonstrate how linear guidance can mitigate this conflict, with the relevant results included in __Section 4, Paragraph “Conflicted Terms.”__
>
> These modifications have been highlighted in __red__ in the revised manuscript.
>
> __For the change request No.3: Rectified Flow Model (see Section 6.2)__
>
> We have added experiments in __Section 6.2__ and qualitative results in __Figure 13 and Appendix Figure 24 (in olive color)__. regarding the performance of rectified flow models on linear and linear-clamp families. The experiments demonstrate that the guidance scheduler is also effective for rectified flow models such as SD3 also supported by the qualitative results (__Figure 13 and Appendix Figure 24 (on page 27)__).
>
> We believe the underlying reason is that, although the generation trajectories are rectified during training, the conflict/erroneous direction of the implicit classifier (i.e., the guidance term discussed in newly added __Section 4 (red highlight) and Appendix Section C “toy example”)__ remains difficult to avoid. This suggests that undershooting the guidance at the early stages is still necessary
> to achieve better generation quality.
>
> These modifications have been highlighted in __olive__ colour in the revised manuscript.

---

### Review · Reviewer_5x2e · 2024-10-08

**Summary Of Contributions:**

The paper systematically studies the classifier-free guidance (CFG) for improving the performance of diffusion models. It provides a comprehensive empirical analysis of CFG through various experiments and a real user subjective study. Based on the observations, they suggest using a simple monotonically increasing hyperparameter-free weight scheduler, that could improve the generation performance with only 1 line of code implementation. In addition, they observed a lack of generalization for other tasks using parameterized schedulers with tuned parameters for one task.

**Audience:**

Yes

**Broader Impact Concerns:**

There might be potential copyright and ownership issues. For example, Pikachu (shown in this paper Figure 7) is a trademarked character owned by Nintendo and The Pokémon Company. Using AI to generate images of Pikachu without proper authorization could potentially infringe on these copyrights and trademarks.

**Claims And Evidence:**

Yes

**Requested Changes:**

1. Including theoretical analysis of the training behavior using different schedulers would be beneficial to this work.
2.  While the proposed simple scheduler improved the performance, it would be interesting to include more discussion about when the proposed simple weight scheduler would fail to work, and provide more guidance on the selection of the scheduler based on different factors such as the model architectures.

**Strengths And Weaknesses:**

Strength:
1. This paper includes a comprehensive analysis of CFG weight schedulers' results to fill the gap in the existing literature, which is helpful for both understanding each scheduler's behavior and further improving the generation performance.
2. The study includes experiments across multiple tasks, models, datasets, and a real user study.
3. The paper explores both complex parameterized methods and simple heuristic approaches and is filled with enough details for experimental setup and results.

Weakness:
1. The paper lacks technical contribution, while focusing on experimental analysis.
2. While the paper provides empirical observations for different CFG schedulers, it is heavily focused on empirical results and doesn't offer a deep theoretical explanation for why these scheduling strategies work. Therefore, the results and observations may not generalize well on other different models and tasks, potentially limiting their broad applicability.
3. Since this paper focuses on empirical observations, then including more negative cases where the simple linear scheduler fails to generalize might be interesting and potentially lead to valuable observations in the future.

---

> ### Author Response · Authors · 2024-10-23
>
> We appreciate the reviewer for the time and effort in reviewing our paper. In response to the reviewer’s concerns, our replies are as follows:
>
> __For the Weakness No.2: Generalisation concern__
>
> We have conducted experiments on four distinct tasks (label-to-image, text-to-image, image-to-image, and a toy example of label-to-Gaussian generation) using five different model backbones (DDPM, LDM, SD1.5, SDXL, and the newly added SD3). These were tested across six different heuristic functions and two families of parameterized functions, with at least three instances of parameters for each family.
> Our observations have remained coherent and consistent across all these combinations. Based on this analysis and the hypotheses regarding the strong performance of guidance, we hope this sufficiently addresses the reviewer’s concern regarding the generalization capacity of our approach.
>
>
> __For the change request No.1: Theoretical Analysis of Training Behavior and Guidance Scheduler Performance (see Section 4, Appendix Section C)__
>
> We interpret the review as referring to the inference process, as classifier-free guidance is not applied during the training phase. To address this, we have added a theoretical justification for classifier-free guidance during the inference process, along with a relevant discussion on the potential reasons for its better performance (see __Section 4, “Conflicted Terms” and Figure 5__).
>
> Specifically, in conjunction with the toy example presented in __Appendix Section C “a toy example”__ in the revised version, we hypothesize that the effectiveness of monotonically increasing guidance (similar to certain parameterized functions) may stem from conflicts within the guidance term (i.e., the gradient of the classifier) during the early stages of the diffusion process. At this stage, the image often lacks sufficient information due to high noise levels, and an excessive guidance scale may cause “shaky” or “curvy” generation trajectories, as illustrated in __Appendix Section C__.
>
> This hypothesis is further supported by our measurements of the conflict level between the generation term and the guidance term. Additionally, we demonstrate how linear guidance can mitigate this conflict, with the relevant results included in __Section 4, Paragraph “Conflicted Terms.”__
>
> These modifications have been highlighted in __red__ in the revised manuscript.
>
> __For the change request No.2: Failure Cases Discussion (see Sections 5.3 and 6.3)__
>
> We have added a detailed discussion with illustrations of some failure cases in __Sections 5.3 and 6.3 with Figure 8__. For parameter-free guidance heuristics, failure often occurs due to overly low guidance at the early stages, which can lead to disrupted structures in the generated images. For parameterized guidance schedulers, the risks are more apparent: incorrectly chosen parameters can degrade overall performance, manifesting as fuzzy details, incomplete structures, or abnormal chromatic information.
>
> These modifications have been highlighted in __red__ in the revised manuscript.
>
> __For the Potential Copyright issue:__
>
> We removed the image of Pikachu to avoid copyright issues, we thank the reviewer for indicating this.

---

### Author Response · Authors · 2024-10-23

We sincerely thank all the reviewers for their thoughtful and valuable feedback on our paper. We are particularly grateful for their recognition of the strength of our experimental contributions (5x2e, wiZb, d2J5), the clarity of our presentation (5x2e, wiZb, d2J5), and the potential impact of our work on the field (d2J5).

We would also like to express our gratitude to the reviewers for raising important concerns and offering constructive feedback. These include requests for deeper insights into Guidance Scheduler performance (5x2e, wiZb), expanding our analysis to include more applications and models such as Stable Diffusion 3 (wiZb) and image-to-image translation (d2J5), and identifying typos, minor errors, and potential copyright issues.

We have incorporated these suggestions into the revised paper (highlighted in red, blue, and olive colours to correspond with specific reviewer requests), and we provide additional explanations for each reviewer separately.

---

### Decision · Action_Editor_WdXJ · 2024-11-17

**Recommendation:** Accept as is

**Comment:**

The paper offers a comprehensive empirical evaluation of CFG weight schedulers, demonstrating their consistent performance improvements with simple, monotonically increasing methods. Reviewers generally agree on its clarity, practical relevance, and thorough experimentation. The authors addressed most concerns, adding theoretical insights and broadening experiments, including rectified flow models and image-to-image tasks. The work is empirically strong and practically useful, and all the reviewers are leaning towards acceptance. Thus, I recommend acceptance.

**Audience:**

general audience who are interested in t2i techniques.

**Claims And Evidence:**

The paper provides a thorough empirical analysis of classifier-free guidance (CFG) weight schedulers in text-to-image diffusion models, highlighting that simple, monotonically increasing schedulers consistently improve performance. While it lacks significant novelty or theoretical depth, its practical recommendations and broad experimental scope are its strengths. The study is relevant for researchers and practitioners but may have limited theoretical support. Reviewers generally lean toward acceptance, praising clarity and utility, though some recommend further exploration of broader applications and theoretical underpinnings.